# Pentatricopeptide repeat poly(A) binding protein KPAF4 stabilizes mitochondrial mRNAs in *Trypanosoma brucei*

Mikhail V. Mesitov[1], Tian Yu[1,2], Takuma Suematsu[1], Francois M. Sement[1], Liye Zhang[3], Clinton Yu[4], Lan Huang[4] & Inna Aphasizheva [1]

In *Trypanosoma brucei*, most mitochondrial mRNAs undergo editing, and 3′ adenylation and uridylation. The internal sequence changes and terminal extensions are coordinated: pre-editing addition of the short (A) tail protects the edited transcript against 3′-5′ degradation, while post-editing A/U-tailing renders mRNA competent for translation. Participation of a poly(A) binding protein (PABP) in coupling of editing and 3′ modification processes has been inferred, but its identity and mechanism of action remained elusive. We report identification of KPAF4, a pentatricopeptide repeat-containing PABP which sequesters the A-tail and impedes mRNA degradation. Conversely, KPAF4 inhibits uridylation of A-tailed transcripts and, therefore, premature A/U-tailing of partially-edited mRNAs. This quality check point likely prevents translation of incompletely edited mRNAs. We also find that RNA editing substrate binding complex (RESC) mediates the interaction between the 5′ end-bound pyr-ophosphohydrolase MERS1 and 3′ end-associated KPAF4 to enable mRNA circularization. This event appears to be critical for edited mRNA stability.

[1] Department of Molecular and Cell Biology, Boston University Medical Campus, Boston, MA 02118, USA. [2] Bioinformatics Program, Boston University, Boston, MA 02215, USA. [3] School of Life Science and Technology, ShanghaiTechUniversity, 201210 Shanghai, China. [4] Department of Physiology and Biophysics, School of Medicine, University of California, Irvine, CA 92697, USA. Correspondence and requests for materials should be addressed to I.A. (email: innaaf@bu.edu)

The hemoflagellate *Trypanosoma brucei* (*T. brucei*) maintains a mitochondrial genome composed of catenated maxicircles and minicircles. A few 23-kb maxicircles encode 9S and 12S rRNAs, six protein-coding and 12 encrypted genes, a *trans*-acting MURF2-II and *cis*-acting CO2 guide RNAs (gRNA). Thousands of 1-kb minicircles produce gRNAs that direct U-insertion/deletion editing of cryptic maxicircle transcripts, thus giving rise to open reading frames[1–3]. Messenger and rRNA precursors are individually transcribed[4] and processed by 3′–5′ exonucleolytic trimming, which is followed by adenylation or uridylation, respectively[5]. Trimming is accomplished by DSS1 3′–5′ exonuclease[6] acting as subunit of the mitochondrial processome (MPsome), which also contains an RNA editing TUTase 1 (RET1) and several structural polypeptides[7]. Binding of the pentatricopeptide repeat (PPR) (35 amino acids) kinetoplast polyadenylation factor 3 (KPAF3) to purine-rich sequences near the encoded 3′ end recruits KPAP1 poly(A) polymerase and channels pre-mRNA into the adenylation/editing pathway[5,8]. Conversely, rRNAs lacking KPAF3-binding sites upstream of the MPsome-generated 3′ end are uridylated by RET1 TUTase[5]. The U-tails decorating ribosomal[9] and guide RNAs[10] reflect a mechanism in which antisense transcripts impede 3′–5′ degradation thereby creating a kinetic window for U-tail addition[5,7]. Thus, uridylation terminates rRNA and gRNA precursor trimming, but the resultant U-tails do not influence the stability of mature molecules[11,12]. In contrast, short A-tails (20–25 nt) exert profound and opposite effects on mRNA decay depending upon the molecule's editing status. Knockdown of KPAP1 poly(A) polymerase leads to moderate upregulation of non-adenylated pre-edited mRNA but causes a rapid degradation of the same transcript edited beyond the initial few sites[5,8,13]. Remarkably, mRNAs containing functional coding sequence that do not require editing, referred to as unedited, also rely on KPAF3 binding and ensuing KPAP1-catalyzed A-tailing for stabilization. In massively edited (pan-edited) transcripts, sequence changes typically begin near the 3′ end and proceed in the 3′–5′ direction[14]. An unknown signaling mechanism monitors editing status and triggers short A-tail extension into a long (>200 nt) A/U-heteropolymer upon completion of the editing process at the 5′ region. The A/U-tailing is accomplished by KPAP1 poly(A) polymerase and RET1 TUTase and requires an accessory heterodimer of PPR proteins KPAF1 and KPAF2. The A/U-tail does not affect the stability, but rather activates mRNA for translation by enabling binding to the small ribosomal subunit[15]. Thus, the temporally separated pre-editing A-tailing and post-editing A/U-tailing processes are distinct in their factor requirements and functions.

Selective KPAF3 binding to G-rich pre-edited, but not to U-rich-edited sequences, likely monitors initiation of mRNA editing at the 3′ end, which rationalizes the editing-dependent stability phenomenon[5]. It follows that KPAF3-bound pre-edited mRNA is protected against 3′–5′ degradation and remains stable while losing A-tail upon KPAP1 knockdown[5,8]. It has been suggested that KPAF3 displacement by the editing sequence changes would leave the partially edited transcript reliant on the short A-tail as an critical stability determinant[5]. This model, however, does not explain the resistance of adenylated RNA to either degradation by the MPsome, or uridylation by RET1 in vivo. Indeed, these features would be essential for partially edited mRNA stabilization and for blocking its A/U-tailing, hence premature translational activation. However, synthetic adenylated RNA represents a susceptible substrate for degradation by the MPsome[7] and uridylation by RET1[16] in vitro.

Recent identification of the 5′ pyrophosphohydrolase complex (PPsome) introduced another dimension to the mRNA processing and stabilization pathway[4]. The PPsome is comprised of three subunits: MERS1, a NUDIX (nucleoside diphosphates linked to x (any moiety)) hydrolase; MERS2 PPR factor; and MERS3, a subunit lacking any motifs. The PPsome binds the 5′ end of a primary transcript and converts the 5′ triphosphate moiety incorporated at transcription initiation into a monophosphate. Intriguingly, MERS1 knockdown severely compromises edited mRNA stability without affecting 3′ polyadenylation. To reconcile these observations, we hypothesized that poly(A)-binding protein (PABP) may inhibit mRNA 3′–5′ degradation and 3′ uridylation by sequestering the short A-tail. We further reasoned that PABP may interact with the PPsome at the 5′ end to stabilize mRNA during the editing process. Unable to identify a canonical RRM motif-containing PABP in mitochondria, we inquired whether a PPR factor capable of recognizing adenosine stretches may exist. A recognition code developed for PPRs from land plants suggests that each repeat binds a single nucleotide via amino acid situated in positions 5 and 35, or the last residue in helix-turn-helix motifs exceeding the canonical length[17]. For example, a combination of threonine and asparagine in positions 5 and 35, respectively, recognizes an adenosine base[18,19].

By searching for repeats with T/N pattern among 38 predicted PPRs[20], we identified a polypeptide containing five adjacent repeats predicted to bind as many contiguous adenosines. Termed kinetoplast polyadenylation factor 4 (KPAF4), this protein interacts with established components of the polyadenylation and editing complexes and predominantly binds to short A-tails in vivo. KPAF4 knockdown downregulates A-tailed edited and unedited mRNAs, but not their A/U-tailed forms. Remarkably, KPAF4 repression also permits uridylation of A-tailed pre-edited mRNAs. Specific KPAF4 binding to adenylated substrate inhibits both 3′–5′ RNA degradation by the MPsome and uridylation by RET1 TUTase in vitro. Collectively, our data support a model in which KPAF4 binding to the short A-tail and interaction with the PPsome enable intramolecular circularization, thus inhibiting mRNA decay by the MPsome.

## Results

**KPAF4 interacts with mitochondrial mRNA processing complexes.** To identify a mitochondrial PABP, we analyzed the repeat structure and amino acids occupying positions 5 and 35, or the last position in repeats longer than 35 residues[17], in annotated PPR-containing polypeptides from *T. brucei*[20]. We searched for threonine and asparagine residues in these positions, respectively, a combination that binds an adenosine[18,19]. By considering proteins with at least four adjacent repeats, we identified a candidate 31.8 kDa protein termed KPAF4 (Tb927.10.10160), which consists almost entirely of seven PPR repeats. The 6–7 repeat organization is conserved among orthologous proteins in *Trypanosoma* and *Leishmania* species, while repeats R1–R5 invariantly possess a T–N combination (Fig. 1a). Repeats 6 and 7 had the required combination shifted by one position. Because topology prediction algorithms ranked the probability of mitochondrial targeting at 20–40%, the KPAF4 localization was confirmed by subcellular fractionation. The C-terminally TAP-tagged[21] KPAF4 was conditionally expressed in insect (procyclic) form of *T. brucei* and demonstrated to have been enriched in the mitochondrial matrix by approximately eight-fold. Partial association with the inner membrane has also been detected (Fig. 1b).

To place the candidate protein into a functional context, KPAF4 was isolated by tandem affinity chromatography (Fig. 1c). Purifications were also conducted from a parental 29–13 cell line[22] and from RNase I-treated mitochondrial lysate. Final fractions were analyzed by immunoblotting for established mRNA processing factors (Fig. 1d). KPAP1, KPAF1, and KPAF3 were readily detectable among proteins co-purifying with KPAF4,

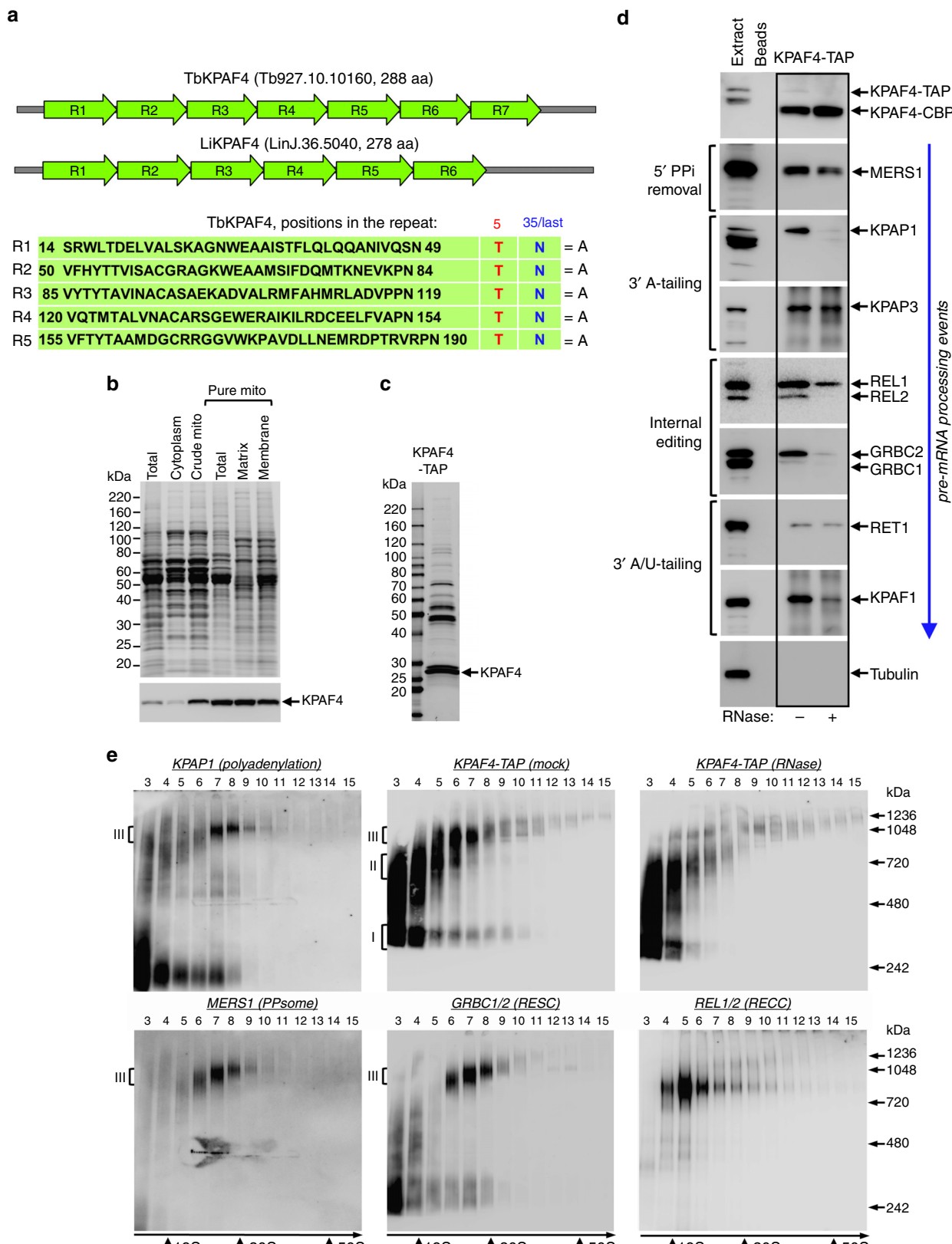

but the KPAP1 and KPAF1 association appeared to be RNA-dependent. RNase treatment also reduced KPAF4 interactions with the PPsome (MERS1[12]), RNA editing core (RECC) (REL1/2[23,24]) and substrate-binding (GRBC1/2[12,25]) complexes, and KPAF1/2 polyadenylation factor[15]. Only a trace amount of RET1 TUTase[26] was detected in the KPAF4 fraction.

Co-purification with protein complexes responsible for mRNA 5′ end modification, editing, A-tailing, and A/U-tailing indicates that KPAF4 likely participates in mRNA processing, and that some interactions are RNA-dependent. To assess the heterogeneity and apparent molecular mass of KPAF4-containing particle(s) in relation to established mRNA processing complexes,

**Fig. 1** Repeat organization, subcellular localization, and complex association of KPAF4. **a** Schematic repeat organization of kinetoplast polyadenylation factor 4 from *Trypanosoma brucei* (Tb) and *Leishmania infantum* (Li). Repeat boundaries were determined using the TPRpred online tool (https://toolkit. tuebingen.mpg.de/#/tools/tprpred) and adjusted according to Cheng et al. [17]. Amino acids in positions 5 and 35/last potentially involved in adenosine recognition are indicated in separate columns. **b** Mitochondrial targeting of KPAF4-TAP fusion protein. Crude mitochondrial fraction was isolated by hypotonic lysis and differential centrifugation (crude mito), and further purified by renografin density gradient (pure mito). The latter preparation was extracted under conditions that separate matrix from membrane-bound proteins[45]. Protein profiles were visualized by Sypro Ruby staining and KPAF4-TAP was detected with an antibody against the calmodulin-binding peptide. The mitochondrial enrichment was calculated by quantitative western blotting vs. total protein loading. Representative of two experiments is shown. **c** Tandem affinity purification of KPAF4. Final fraction was separated on 8–16% SDS gel and stained with Sypro Ruby. Representative of three experiments is shown. **d** KPAF4 co-purification with mRNA processing complexes. Fractions purified from parental cell line (beads, no tagged protein expressed), and mock and RNase-treated mitochondrial extracts were subjected to immunoblotting with antibodies against MERS1 NUDIX hydrolase (PPsome subunit), KPAP1 poly(A) polymerase, KPAF1 and KPAF3 polyadenylation factors, and GRBC1/2 (RNA editing substrate-binding complex, RESC) and RET1 TUTase (MPsome). Tagged KPAF4 was detected with antibody against calmodulin-binding peptide. RNA editing core complex (RECC) was detected by self-adenylation of REL1 and REL2 RNA ligases in the presence of [$\alpha$-$^{32}$P]ATP. Representative of two experiments is shown. **e** Crude mitochondrial fraction was extracted with detergent and soluble contents were separated for 5 h at 178,000×*g* in a 10–30% glycerol gradient. Each fraction was resolved on 3–12% Bis–Tris native gel. Positions of native protein standards are denoted by arrows. KPAP1, KPAF4-TAP, MERS1, and GRBC1/2 were visualized by immunoblotting. REL1 and REL2 RNA ligases were detected by self-adenylation. Thyroglobulin (19S) and bacterial ribosomal subunits were used as apparent *S*-value standards. In each panel, representative of three experiments is shown

mitochondrial lysates from parental and KPAF4-TAP cells were fractionated on glycerol gradients. Fractions were separated on a native gel and analyzed for polyadenylation, PPsome, RNA editing core (RECC), and substrate-binding (RESC) complexes (Fig. 1e). In agreement with previous studies, KPAP1 was detected in an unassociated form and bound to an ~1 MDa complex[5,8], while KPAF4 was separated into particles of ~300 kDa (I) and ~600 kDa (II), and attached to an ~1 MDa complex (III, fractions 6 and 7). Notably, RNase pre-treatment of mitochondrial lysate mostly eliminated the 1 MDa KPAF4 complex III but left smaller particles unaffected. The PPsome and RNA editing substrate binding complex (RESC) co-fractionated as an ~1 MDa particle that closely resembles complex III, while the RECC migrated as a distinct ~800 kDa particle. Collectively, these results demonstrate that KPAF4 is a mitochondrial PPR factor engaged with at least three macro-molecular complexes. The largest KPAF4-containing complex III with an apparent molecular mass of ~1 MDa closely resembles a ribonucleoprotein assembly that encompasses PPsome, RESC, and polyadenylation complexes[4,25].

**RESC tethers PPsome and polyadenylation complexes**. To gain a higher-resolution view of the KPAF4 interactome, the nor-malized spectral abundance factors (NSAF)[27] were derived from LC–MS/MS analysis of tandem affinity purified complexes and used to build an interaction network (Fig. 2a). Polyadenylation enzyme KPAP1 and factors KPAF1, KPAF2, and KPAF3 were analyzed along with the MERS1 subunit of the PPsome[5,8,25]. The strongest predicted KPAF4 interactions included those with a hypothetical protein lacking discernible motifs, Tb927.3.2670, and with the polyadenylation-mediating module (PAMC) of the RESC[25]. KPAP1 and KPAF3 also featured prominently among KPAF4-associated proteins. Interestingly, relatively high levels of MRP1 and MRP2 were detected in KPAF4 preparation (Supple-mentary Data 1). A heterotetramer MRP1/2 chaperone displays RNA annealing activity in vitro, but its definitive function remains undetermined[28–31], albeit contribution to mRNA stabi-lity seems likely[32]. The interactions between KPAF4, Tb927.3.2670, and the MRP1/2 RNA chaperone complex have been verified by cross-tagging of MRP2 and the hypothetical protein. Mass spectrometry analysis of samples purified from RNase-treated extracts indicated that interactions between KPAP1 poly(A) polymerase, and KPAF1-2 and KPAF3 poly-adenylation factors are sufficiently stable to withstand a two-step purification, but nonetheless depend on an RNA component

(Supplementary Data 1). KPAF4–MRP1/2–Tb927.3.2670 co-purification, on the other hand, was unaffected by RNase treat-ment. Importantly, the network predicted that the RESC complex may facilitate co-complex contacts between the PPsome and KPAF4.

To corroborate the interaction network inferences, we investigated the proximity of KPAF4, polyadenylation, RESC, and PPsome complexes by in vivo biotinylation (BioID[33]), which has an estimated 10 nm labeling range[34]. KPAP1, GRBC2, MERS1, and KPAF4 were conditionally expressed as C-terminal fusions with BirA* biotin ligase and biotinylation was induced for 24 h. Labeled proteins were purified and analyzed by LC–MS/MS (Fig. 2b and Supplementary Data 2). The BioID experiments placed KPAP1 in proximity to the KPAF2 polyadenylation factor, subunits P3 and P4 of the polyadenylation mediator module (PAMC), and Tb927.3.2670. Surprisingly, MRP2 emerged as the major biotinylated protein in cells expressing KPAP1, MERS1, and GRBC2 fusions with BirA*. In aggregate, the co-purification, apparent molecular mass assessment of KPAF4 complexes and in vivo proximity studies suggest that KPAF4 interacts with the mitochondrial polyadenylation and RESC. It seems plausible that GRBC and REMC modules of the latter mediate the co-complex interaction between KPAF4 and the PPsome.

**KPAF4 is essential for normal parasite growth and for main-taining a subset of mitochondrial mRNAs**. The potential role of KPAF4 in mitochondrial RNA processing and parasite viability was examined in the insect (procyclic) form of *T. brucei*. Indu-cible RNAi knockdown efficiently downregulated KPAF4 mRNA (Fig. 3a) and triggered a cell growth inhibition phenotype after ~72 h, indicating that KPAF4 is essential for normal cellular function (Fig. 3b). Quantitative RT-PCR of RNA samples isolated at 55 h post-RNAi induction demonstrated divergent effects of KPAF4 knockdown on mRNA abundance. Downregulation of moderately edited (CYB and MURF2), and some pan-edited (RPS12, ND3, and CO3) mRNAs was accompanied by upregu-lation of their respective pre-edited forms. The transcript-specific effects were also apparent for unedited transcripts that either remained relatively steady (CO1 and ND5) or increased (ND1, MURF1, and ND4). Finally, mitochondrial ribosomal RNAs remained virtually unaffected, which indicates an mRNA-specific KPAF4 function (Fig. 3c). We next tested whether these effects may have been caused by KPAF4 RNAi-induced changes in steady-state levels of known processing factors. Immunoblotting analysis showed that KPAP1 poly(A) polymerase was

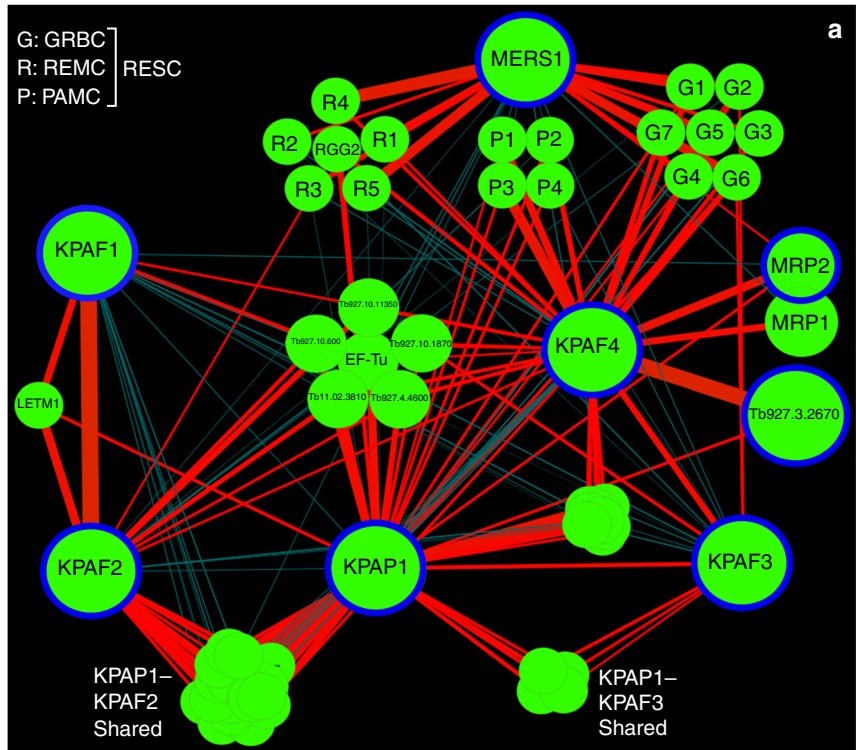

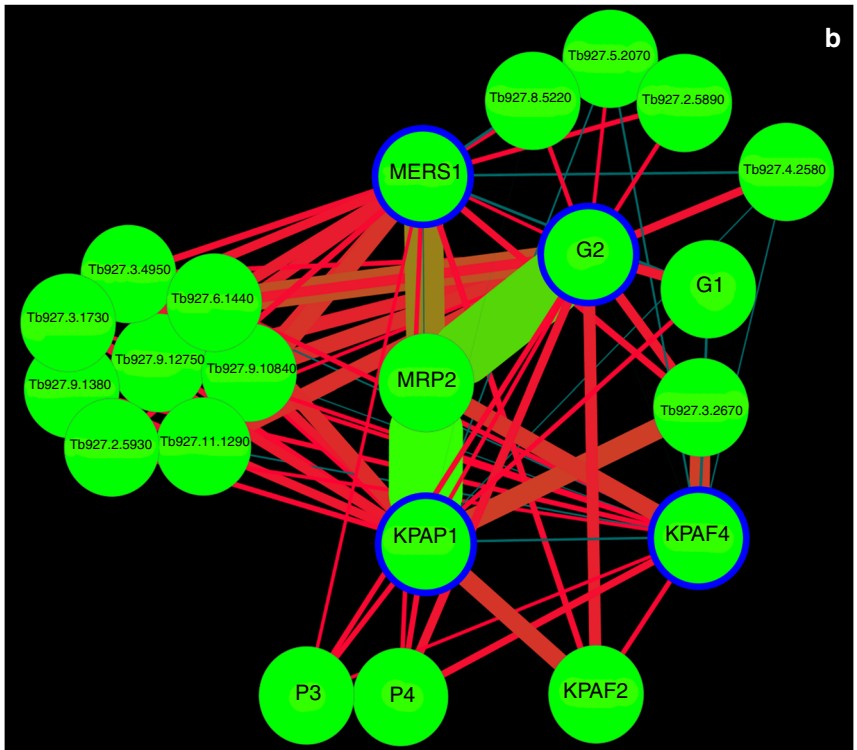

**Fig. 2** KPAF4 interactions and proximity networks. **a** Model of the interactions between KPAF4, KPAP1 poly(A) polymerase, KPAF1-2, and KPAF3 polyadenylation factors, RNA editing substrate-binding complex (RESC), and MRP1/2 RNA chaperones. KPAP1, KPAF1, KPAF2, KPAF3, KPAF4, MRP2, MERS1, and Tb927.3.2670 proteins (encircled in blue) were affinity purified from mitochondrial lysates. The network was generated in Cytoscape software from bait–prey pairs in which the prey protein was identified by five or more unique peptides. The edge thickness correlates with normalized spectral abundance factor (NSAF) values ranging from $2.9 \times 10^{-3}$ to $4.4 \times 10^{-5}$ (Supplementary Data 1). Edges between tightly bound RESC modules (GRBC, REMC, and PAMC) were omitted for clarity[25]. All purifications were performed in parallel under uniform conditions. **b** KPAF4 proximity network. Spectral counts derived from BioID experiments with KPAP1, KPAF4, GRBC2, and MERS1 fusions with BirA* biotin ligase (encircled in blue) were processed as in (**a**) to build a proximity network. The edge thickness correlates with normalized spectral abundance factor (NSAF) values ranging from $2.9 \times 10^{-3}$ to $2.6 \times 10^{-5}$ (Supplementary Data 2). Edge colors other than red are for visualization purposes only. All purifications were performed in parallel under uniform conditions

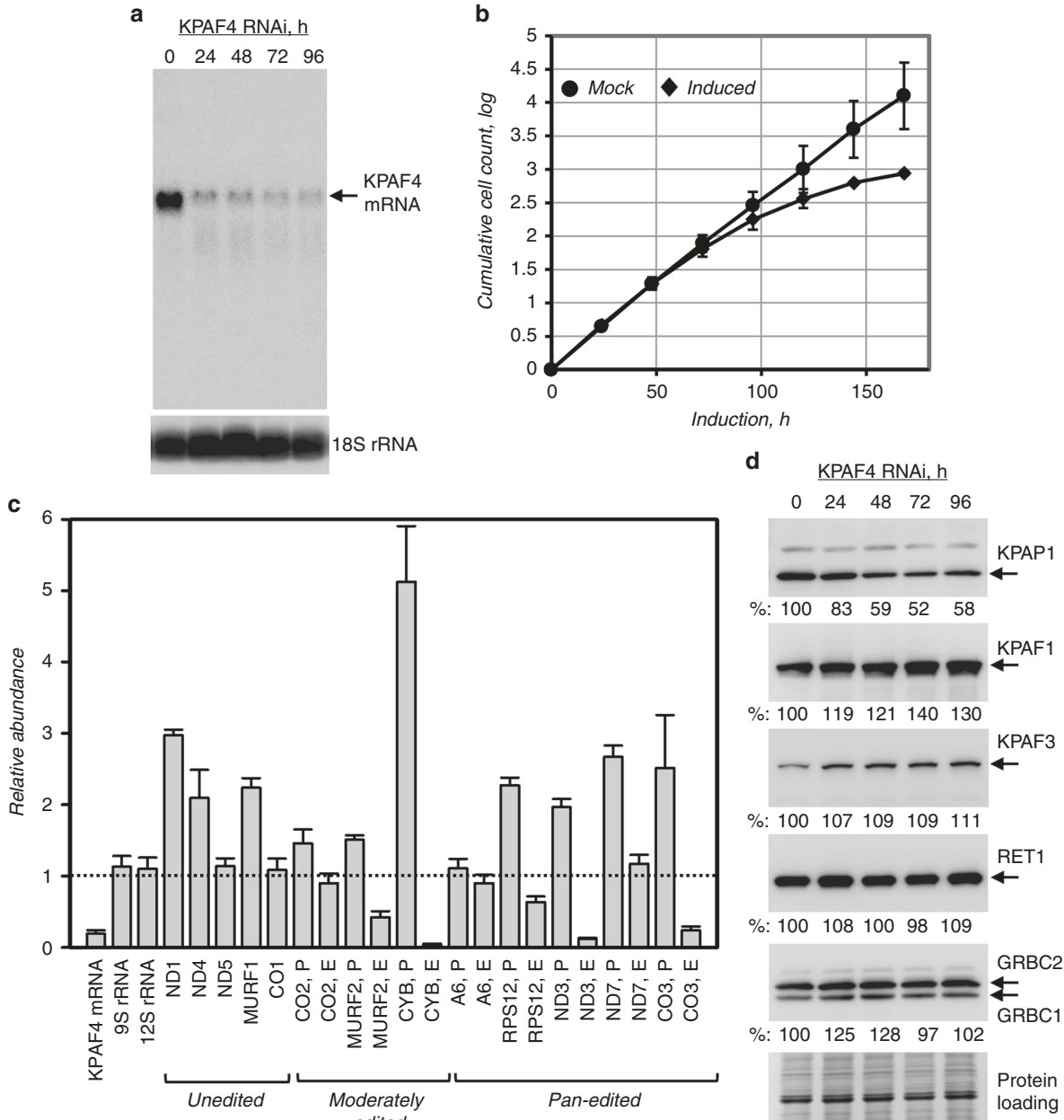

**Fig. 3** KPAF4 repression effects on cell growth and polyadenylation complex. **a** Northern blotting analysis of KPAF4 mRNA downregulation by inducible RNAi. **b** Growth kinetics of procyclic parasite cultures after mock treatment and KPAF4 RNAi induction with tetracycline. Data representative of three independent experiments are shown as mean ± s.d. **c** Quantitative real-time RT-PCR analysis of RNAi-targeted KPAF4 mRNA, and mitochondrial rRNAs and mRNAs. The assay distinguishes edited and corresponding pre-edited transcripts, and unedited mRNAs. RNA levels were normalized to β-tubulin mRNA. RNAi was induced for 55 h. Error bars represent the standard deviation from at least three biological replicates. The thick line at "1" reflects no change in relative abundance; bars above or below represent an increase or decrease, respectively. P, pre-edited mRNA; E, edited mRNA. **d** Cell lysates prepared at indicated time points of KPAF4 RNAi induction were sequentially probed and re-probed by quantitative immunoblotting on the same membrane in the following order: antigen-purified rabbit polyclonal antibodies against KPAP1, KPAF1, KPAF3, GRBC1/2, and mouse monoclonal antibodies against RET1 TUTase. Signals were normalized against three independent loading standards and mean values calculated

downregulated by ~50% in KPAF4 RNAi background while other enzymes and RNA-binding proteins remained unchanged (Fig. 3d).

**KPAF4 knockdown differentially affects mRNAs depending on their editing status.** Albeit instructive, the global changes in relative abundance provide limited information about 3′ modifications and their correlation with mRNA editing status. To assess whether moderate KPAP1 decline in the KPAF4 RNAi

background (Fig. 3d) may have compromised mRNA adenylation, we performed time-resolved analysis of pan-edited mRNAs. The representative example, RPS12 mRNA, constitutes a single domain in which editing initiates close to the polyadenylation site and traverses the entire transcript in a 3′–5′ hierarchical order dictated by overlapping gRNAs[14]. Samples from KPAF3 knockdown were analyzed alongside to typify impeded mRNA adenylation and accelerated decay[5]. Northern blotting of pre-edited, partially-edited (~70% completed, 5′ region not edited) and fully edited variants also distinguishes non-adenylated, A-tailed, and

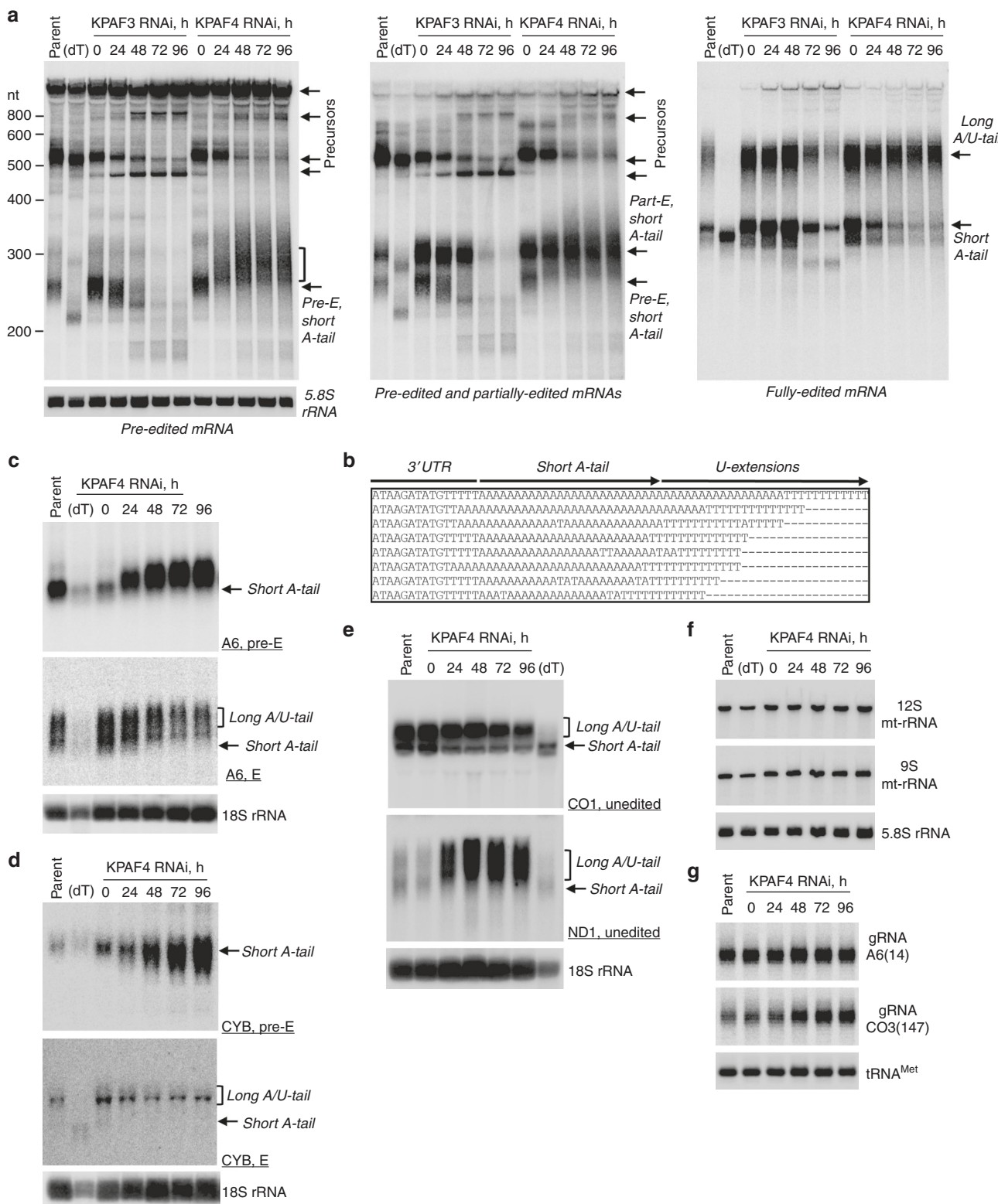

A/U-tailed mRNAs (Fig. 4a, Supplementary Fig. 1). Upon KPAF3 repression, an initial loss of the short A-tail (0–48 h of RNAi induction), was followed by rapid mRNA degradation. In contrast, KPAF4 knockdown led to lengthening and, in agreement with qRT-PCR results (Fig. 3c), to a moderate increase in pre-edited mRNA abundance. While partially edited mRNA patterns mirrored the loss of the pre-edited form in KPAF3 knockdown, similar populations remained virtually unchanged with

progression of KPAF4 RNAi. The fully edited transcripts displayed a more complex pattern: The A-tailed form declined while the A/U-tailed form remained unaffected. To investigate the unexpected lengthening of pre-edited RNAs in KPAF4 knockdown cells, the 3′ extensions were amplified, cloned, and sequenced. In agreement with a previous report for the parental 29-13 strain of *T. brucei*[8], in 96 clones obtained from mock-induced KPAF4 RNAi short A-tails varied within 20–25 nt range.

**Fig. 4** Divergent effects of KPAF4 knockdown on mitochondrial RNAs. **a** Northern blotting of pre-edited (Pre-E), partially edited (Part-E), and fully edited RPS12 mRNA variants. Total RNA was separated on a 5% polyacrylamide/8 M urea gel and sequentially hybridized with radiolabeled single-stranded DNA probes. Zero-time point: mock-induced RNAi cell line. Cytosolic 5.8S rRNA was used as loading control. Parent, RNA from parental 29-13 cell line; (dT), RNA was hybridized with 20-mer oligo(dT) and treated with RNase H to show positions of non-adenylated molecules in parental cell line. Pre-edited RNA length increase in KPAF4 RNAi is shown by brackets. Representative of four experiments for edited and three experiments for pre-edited RPS12 mRNA forms are shown. **b** Alignment of representative RPS12 mRNA 3′ ends in KPAF4 RNAi cells. RNA termini were amplified by cRT-PCR, cloned and sequenced[8]. A fragment of 3′ untranslated region, short A-tail, and U-extensions are indicated. **c** Northern blotting of pan-edited A6 mRNA. Total RNA was separated on a 1.7% agarose/formaldehyde gel and sequentially hybridized with oligonucleotide probes for pre-edited and fully edited sequences. Loading control: cytosolic 18S rRNA. Representative of three experiments is shown. **d** Northern blotting of moderately edited *cyb* mRNA. Total RNA was separated on a 1.7% agarose/formaldehyde gel and hybridized with oligonucleotide probes for pre-edited and fully edited sequences. Loading control: cytosolic 18S rRNA. Representative of two experiments is shown. **e** Northern blotting of unedited CO1 and ND1 mRNAs. Total RNA was separated on a 1.7% agarose/formaldehyde gel and sequentially hybridized with oligonucleotide probes. Loading control: cytosolic 18S rRNA. Representative of two experiments is shown. **f** Northern blotting of mitochondrial ribosomal RNAs. Total RNA was separated on a 5% polyacrylamide/8 M urea gels and hybridized with oligonucleotide probes. Loading control: cytosolic 5.8S rRNA. Representative of two experiments is shown. **g** Guide RNA northern blotting. Total RNA was separated on a 10% polyacrylamide/8 M urea gel and hybridized with oligonucleotide probes specific for gA6(14) and gCO3(147). Mitochondrially localized tRNA$^{Cys}$ served as loading control. Single experiment performed

Remarkably, A-tails not only persisted in KPAF4 knockdown, but in ~30% of clones were extended into oligo(U) stretches (Fig. 4b and Supplementary Data 3). These results demonstrate that, unlike KPAF3, KPAF4 is not required for pre-edited mRNA stabilization and adenylation, but it may prevent spurious uridylation of A-tailed transcripts. The disposition apparently changes with progression of editing in KPAF4 RNAi background: Fully edited short A-tailed mRNA declines while A/U-tailed transcript remains unaffected. It follows that KPAF4 may stabilize fully edited A-tailed mRNA but is not required for its A/U-tailing upon completion of editing.

Extending northern blotting analysis to another pan-edited mRNA encoding subunit A6 of the ATP synthase showed a similar response to KPAF4 depletion: lengthening and upregulation of pre-edited RNA accompanied by downregulation of the edited A-tailed form (Fig. 4c). In moderately edited CYB mRNA, where 34 uridines are inserted close to the 5′ end, the pre-edited form was upregulated while the edited variant behaved like pan-edited mRNAs (Fig. 4d). In unedited mRNAs, such as CO1 and ND1, short A-tailed populations also declined while A/U-tailed ND1 increased more than 10-fold (Fig. 4e, Supplementary Fig. 1). Finally, the lack of detectable impact on rRNAs (Fig. 4f), which are also produced from maxicircle and normally uridylated, confirmed that KPAF4 is an mRNA-specific factor. Minicircle-derived gRNAs were either unaffected, such as gA6(14), or moderately upregulated, as in the case of gCO3(147) (Fig. 4g). The latter effect correlates with a loss of corresponding edited CO3 mRNA (Fig. 3c), as reported for genetic knockdowns that eliminate edited mRNAs[11]. Thus, the outcomes of KPAF4 knockdown are consistent with a hypothetical function of PABP: stabilization of A-tailed edited mRNA that is no longer bound by KPAF3[5], but not yet channeled into the post-editing A/U-tailing reaction[15].

**KPAF4 inhibits mRNA uridylation in vivo**. In pre-edited mRNA, the mature 3′ end is produced by MPsome-catalyzed trimming and KPAF3-stimulated adenylation[5]. The short A-tailed mRNA is then somehow protected from 3′–5′ degradation during editing, and from KPAF1/2-stimulated A/U-tailing[15] until the editing process is completed[8]. Although conventional sequencing provided preliminary indication that KPAF4 may inhibit uridylation of A-tailed mRNA (Fig. 4b), this technique's limitations prevented analysis of longer A-rich extensions. To obtain a comprehensive view of mRNA 3′ termini in KPAF4 RNAi background, we combined mRNA circularization with single molecule real-time sequencing (SMRT, PacBio platform)

and sequencing-by-synthesis (Illumina platform) to characterize short and long tails in pre-edited, edited, and unedited mRNAs, and ribosomal RNAs (Supplementary Fig. 2). RNAs expected have only short tails, such as pre-edited RPS12, A6, and CYB transcripts, and 12S rRNA, were sequenced on Illumina platform while their edited forms known to have both short and long tails were sequenced with PacBio platform[35]. Unedited CO1 mRNA, expected to have short and long tails, and U-tailed rRNA was sequenced on both platforms. The long-range SMRT sequencing of A/U-tails revealed an ~50:50 A/U ratio in edited and unedited mRNAs (Fig. 5d), which is somewhat different than previously calculated 70:30 ratio[15]. The molecular cloning of 3′ extensions in the original report likely caused the differences with this study. Length classification of short 3′ extensions into 10 nt bins (Fig. 5a), and long ones into 10 nt and 50 nt bins (Fig. 5b), exposed higher heterogeneity and general shortening of short A-tails in pre-edited transcripts upon KPAF4 RNAi induction for 72 h. In contrast, corresponding pan-edited RPS12 and A6 mRNAs, and unedited CO1, possessed a higher percentage of tails in the 150–250 nt range, which encompasses the bulk of A/U-tailed mRNAs. The lack of effect on rRNAs further establishes KPAF4 as an mRNA-specific factor (Fig. 5a, c). We also noticed that the A/U-tail length distribution derived from real-time sequencing was consistent with the apparent length determined by northern blotting (Fig. 4a), as sequences longer than 400 nt were detected (Fig. 5d). Plotting of nucleotide frequencies from short-range sequencing also confirmed A-tail shortening accompanied by uridylation in pre-edited RPS12 and A6 mRNAs (Fig. 5c). As indicated by distribution of adenosine and uridine residues in long tails, lack of KPAF4 leads to earlier emergence of U-rich structures (Fig. 5d). Noteworthy, the short-range Illumina sequencing confirmed rRNA's uridylated status, while the real-time technique was uninformative for short 3′ extensions. In conclusion, 3′ tail sequencing on two independent platforms connected the loss of KPAF4 with the spurious addition of U-tails to adenylated mRNAs, and with general stimulation of A/U-tail synthesis.

**KPAF4 binds A-tails in vivo**. To establish KPAF4 in vivo-binding sites, we applied UV-crosslinking of live cells, and two-step affinity purification of TAP-6His-tagged polypeptide followed by deep sequencing (CLAP-Seq, Fig. 6a). We note that maxicircle genes encode rRNAs, and unedited and pre-edited mRNAs, which are typically separated by short non-coding regions. Since most genes are individually transcribed as 3′ extended precursors, the mature mRNA 3′ ends produced by

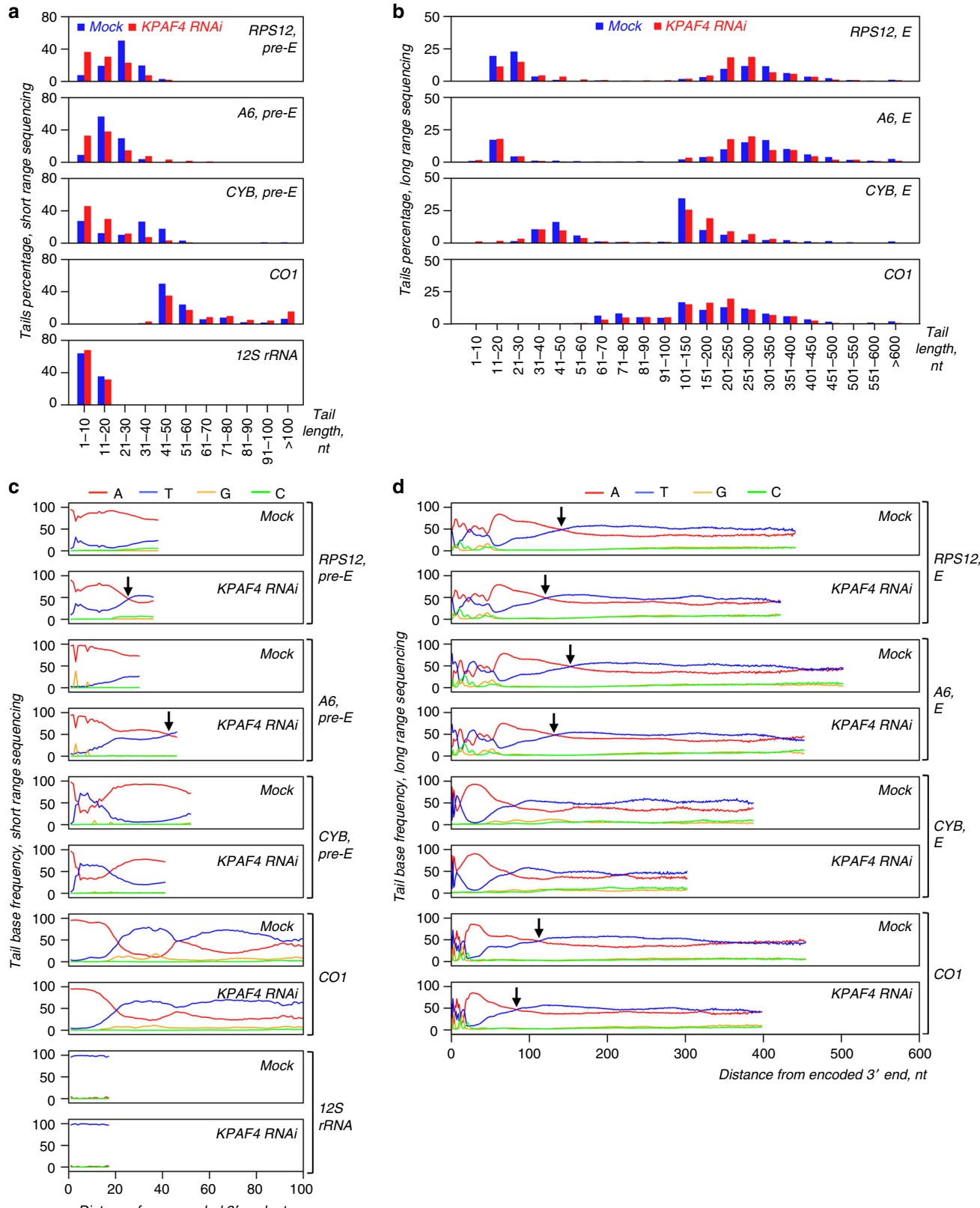

3′–5′ trimming often extend into 5′ regions of downstream genes[4]. In KPAF4 CLAP-Seq, ~40 × 10⁶ reads originated from maxicircle transcripts and edited mRNAs, while only ~9 × 10⁶ reads mapped to the minicircles constituting more than 90% of kinetoplast DNA[36,37]. Mapping of CLAP-Seq reads to the maxicircle revealed a preference for 3′ ends of pre-edited and unedited transcripts encoded on both strands. Conversely, most reads derived from abundant rRNAs clustered within 9S rRNA (Fig. 6b). At the mRNA level, plotting a nucleotide frequency within reads that partially mapped to unedited and edited transcripts demonstrated a strong bias toward adenosine residues at the 3′ region (Fig. 6c). A composite read mapping and nucleotide

**Fig. 5** Sequencing of mRNA and rRNA 3′ extensions in KPAF4 RNAi background. **a** Length distribution of short mRNA and 12S rRNA tails. Non-encoded 3′ end extensions (MiSeq instrument, Illumina, single biological replicate) were individually binned into 10-nt length groups. Mock-induced and RNAi datasets, indicated by blue and red bars, respectively, represent percentage of the total number of reads. **b** Length distribution of long mRNA tails. Non-encoded 3′ end extensions (PacBio RS II instrument, two biological replicates) were individually binned into 10-nt length groups before 100 nt, and in 50-nt groups thereafter. Mock-induced and RNAi datasets are indicated by blue and red bars, respectively, that represent percentage of the total number of reads. **c** Positional nucleotide frequencies in short mRNAs and 12S rRNA tails. A nucleotide percentage was calculated for each position that contained at least 5% of the total extracted sequences. The nucleotide bases are color-coded as indicated. Arrows show positions of equal adenosine and uridine frequencies. **d** Positional nucleotide frequencies in long mRNA tails. A nucleotide percentage was calculated for each position that contained at least 5% of the total extracted sequences. The nucleotide bases are color-coded as indicated. Arrows show positions of equal adenosine and uridine frequencies

frequency plot calculated for unedited and fully edited mRNAs with the termination codon set as zero further demonstrates KPAF4's preferential binding near polyadenylation sites and to short A-tails, but not long A/U-tails (Fig. 6d). Interestingly, pure A-tracks accounted for ~0.5% ($2 \times 10^5$) of all unmapped KPAF4-CLAP reads while fragments ending with more than 30 As constituted 33% ($1.5 \times 10^6$) of all reads mapped to mitochondrial mRNAs. Tail sequencing and KPAF4-CLAP statistics are provided in Supplementary Data 4.

To test whether in vivo poly(A)-binding specificity is conferred by amino acid residues occupying positions 5 and 35 or the last residue in KPAF4 repeats, we introduced T5N and N35/36D substitutions into all seven PPRs (Fig. 1a). The expression levels of mutated variant (KPAF4-Mut) and KPAF4-WT were virtually identical (Supplementary Fig. 3a) while LC–MS/MS analysis demonstrated a similar composition of respective affinity purified samples (Supplementary Data 5). However, in CLAP-Seq experiments KPAF4-Mut showed markedly reduced crosslinking efficiency (Supplementary Fig. 3b) and background coverage of mitochondrial transcripts (Supplementary Fig. 3c).

KPAF4 knockdown leads to uridylation and upregulation of pre-edited mRNA, but also causes concurrent decay of the A-tailed edited form (Fig. 4a, b, e). To elucidate the connection between the mRNA's editing status and KPAF4-dependent stabilization, we compared read coverage between individual pre-edited and fully edited mRNAs; nucleotide frequencies were also included to detect non-encoded 3′ additions (Fig. 6e). A consistent pattern in pan-edited RPS12 and A6 mRNA showed that KPAF4 preferentially binds to the 5′ and 3′ regions, including A-tails, in pre-edited transcripts, but is confined to 3′ regions in fully edited mRNAs. In moderately edited CYB mRNA, the editing-dependent re-distribution of reads was similar, except for adenosine enrichment at the pre-edited 5′ end, a likely outcome of reads mapping to the 3′ end of the closely spaced upstream CO3 mRNA. These observations suggest that KPAF4 binds to both 5′ and 3′ termini in pre-edited transcripts, possibly leading to mRNA circularization. Furthermore, sequence changes introduced by editing and/or remodeling of ribonucleoprotein complexes during the editing process, apparently displace KPAF4 from 5′ regions, where the editing process comes to completion. The circularization suggested by KPAF4 binding to both mRNA ends (Fig. 6e) and cross-talk between 3′ end-bound KPAF4 and 5′ end-bound PPsome (Fig. 2) may be critical for inhibiting 3′–5′ degradation[5]. These observations may provide a mechanistic basis for the rapid decay of edited mRNA in MERS1 knockdown[12]. MERS1 pyrophosphohydrolase binds to the 5′ terminus and removes pyrophosphate from the first nucleotide incorporated by transcription, but the mechanism of mRNA stabilization by MERS1 remains unclear[4]. If circularization indeed takes place, we reasoned that MERS1 would also be expected to bind the 3′ end and/or A-tails. Mapping of MERS1-CLAP reads to the same transcripts exposed the KPAF4-like re-distribution of MERS1-binding sites from the 5′ end in pre-edited to both 5′ and 3′

termini including A-tails in edited mRNAs (Fig. 6f). In sum, in vivo crosslinking experiments indicate that pan-editing events eliminate KPAF4-binding sites in pre-edited transcripts and confine this factor to the 3′ region and short A-tail. These events are likely responsible for KPAF4-mediated protection of A-tailed edited mRNA against 3′–5′ degradation by the MPsome.

**KPAF4 inhibits uridylation and degradation of adenylated RNAs in vitro.** Recombinant KPAP1 poly(A) polymerase activity is intrinsically limited to adding 20–25 adenosines[8], while RET1 TUTase processively polymerizes hundreds of uridines in vitro[26]. Although both enzymes lack a pronounced RNA specificity, RET1 is most efficient on substrates terminating with several Us[16]. Likewise, uridylated RNAs represent the preferred substrate for the MPsome in vivo and in vitro[7]. It follows that a factor responsible for blocking uridylation and stabilization of adenylated mRNA would specifically bind A-tailed RNA and interfere with RET1 and MPsome activities. To investigate whether KPAF4 possesses such properties, we have established an in vitro reconstitution system composed of affinity purified KPAF4 and DSS1 exonuclease complexes, and recombinant KPAP1 and RET1 enzymes. We used synthetic 81 nt RNA resembling a 3′ region of edited RPS12 mRNA, and RNAs extended with either 20 As or 20 Us, in experiments with purified KPAF4-WT and KPAF4-Mut (Fig. 7a and Supplementary Data 5).

In an electrophoretic mobility shift assay (EMSA), only adenylated RNA formed a single distinct ribonucleoprotein complex commensurate with increasing KPAF4-WT concentration (Fig. 7b). Conversely, KPAF4-Mut failed to bind any of the substrates within the concentration range afforded by the assay (Fig. 7c). In enzymatic reactions with no-tail RNA, RET1 and KPAP1 produced patterns like those reported for generic RNA substrates: distributive addition of ~15 As and processive polymerization of hundreds of Us, respectively (Fig. 7d)[8,38]. In reactions containing a mixture of both enzymes, the extension patterns were dominated by RET1 activity. Uridylated RNA was efficiently utilized by RET1 but proved to be a poor substrate for KPAP1. In contrast to no-tail and U-tailed RNA, KPAP1 inhibited processive uridylation of the A-tailed substrate by RET1 TUTase. Unlike KPAF3, which dramatically stimulates KPAP1 activity on any tested RNA[5], KPAF4 did not produce noticeable effects on either RET1 or KPAP1 activities with no-tail or U-tail RNA. However, KPAF4 inhibited processive uridylation of A-tailed RNA by RET1 TUTase, and this effect was further enhanced by KPAP1. Together, these results demonstrate that KPAF4 specifically recognizes adenylated RNAs and inhibits their uridylation by RET1 TUTase. Importantly, KPAF4's inhibitory effect on uridylation is enhanced by KPAP1 poly(A) polymerase.

The MPsome-catalyzed 3′–5′ degradation represents a major processing pathway for rRNA, mRNA, and gRNA precursors, and is also responsible for decay of mature molecules[5,7]. While KPAF3 has been shown to protect any RNA against degradation

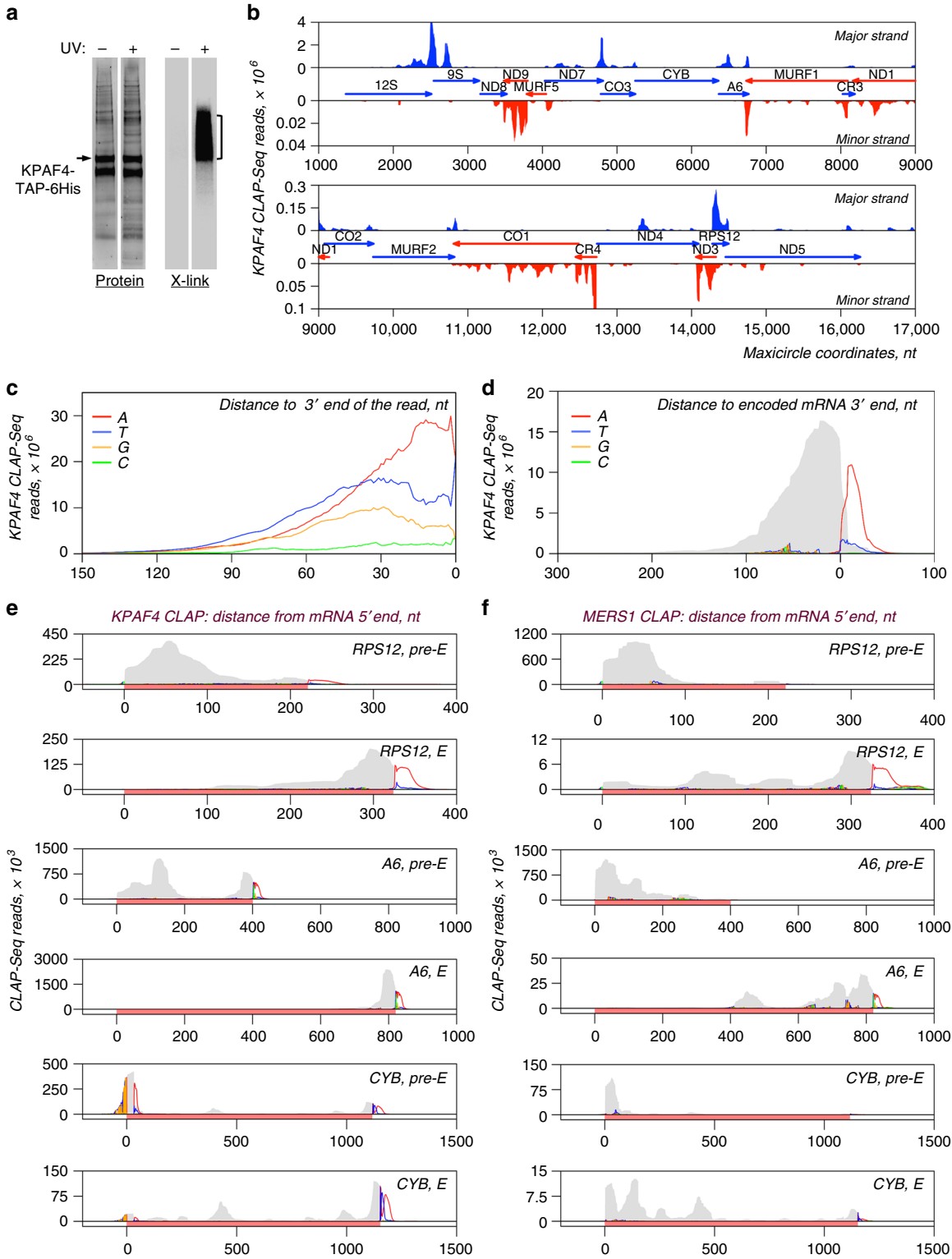

by the MPsome in vitro[5], KPAF4-binding properties and knockdown outcomes suggest that it may preferentially inhibit degradation of adenylated RNAs. To test this hypothesis, we reconstituted mRNA degradation with affinity-purified MPsome and the same 5′ radiolabeled substrates used in binding and 3′ extension assays. Reactions were performed for a fixed duration in the presence of increasing KPAF4 concentrations (Fig. 7e, left

panels), or a time course was followed in the presence of a constant KPAF4 amount (Fig. 7e, right panels). Quantitation of KPAF4 concentration-dependent or time-dependent decrease of input substrate demonstrated that the MPsome degrades no-tail or uridylated RNAs irrespective of KPAF4 presence. However, KPAF4 specifically inhibits hydrolysis of adenylated RNA by the MPsome (Supplementary Fig. 4). These experiments illustrate

**Fig. 6** Distribution of KPAF4 in vivo-binding sites between pre-edited and edited mRNAs. **a** Isolation of in vivo KPAF4-RNA crosslinks. Modified TAP-tagged fusion protein was purified by tandem affinity pulldown from UV-irradiated (+) or mock-treated (−) parasites. The second purification step was performed under fully denaturing conditions and resultant fractions were subjected to partial on-beads RNase I digestion and radiolabeling. Upon separation on SDS–PAGE, RNA–protein crosslinks were transferred onto nitrocellulose membrane. Protein patterns were visualized by Sypro Ruby staining (left panel), and RNA–protein crosslinks were detected by exposure to phosphor storage screen (right panel). RNA from areas indicated by brackets was sequenced. Representative of six biological replicates is shown. **b** KPAF4 in vivo-binding sites. Crosslinked fragments were mapped to the maxicircle's gene-containing region. Annotated mitochondrial transcripts encoded on major and minor strands are indicated by blue and red arrows, respectively. **c** Position-specific nucleotide frequency in partially mapped KPAF4 CLAP-Seq reads. In reads selected by partial mapping to maxicircle and edited mRNAs, the unmapped 3′ segments were considered as tail sequences. The nucleotide frequency was calculated for each position beginning from the 3′ end. **d** Aggregate KPAF4 mRNA-binding pattern. Read coverage is represented by the gray area, and the nucleotides in 3′ extensions are color-coded at their projected positions. **e** KPAF4 binding to representative pan-edited (RPS12, A6) and moderately edited (CYB) mRNAs. Read coverage profiles were created for matching pre-edited and fully edited mRNA. Read coverage is represented by the gray area, and the unmapped nucleotides in 3′ extensions are color-coded at their projected positions. The mRNA is highlighted with a rose bar in the context of adjacent maxicircle sequences. **f** MERS1 binding to representative pan-edited (RPS12, A6), and moderately edited (CYB) mRNAs. Graphs were created as in panel **e**

that KPAF4 in vitro properties are consistent with the expected functions of a PABP in: (1) recognizing the A-tail; (2) protecting adenylated mRNA against premature uridylation by RET1 TUTase; and (3) inhibiting degradation of adenylated mRNA by the MPsome.

## Discussion

Studies of the unicellular parasite *T. brucei* revealed physical interactions and functional coupling between protein complexes that convert cryptic mitochondrial transcripts into translation-competent mRNAs. Among many transformations, constrained adenylation by KPAP1 poly(A) polymerase is critical for edited and unedited mRNA stability[8,13]. Addition of 20–25 adenosines is stimulated by KPAF3 polyadenylation factor, which is recruited to pre-edited mRNA, but is then displaced by editing events[5]. Thus, transcripts edited beyond a few initial sites depend on the short A-tail for protection against destruction by the mitochondrial processome. Although 3′–5′ degradation is the main decay mechanism, mRNA stabilization also requires binding of PPsome subunit MERS1 to the 5′ end. Finally, post-editing A/U-tailing involving RET1 TUTase activates ribosome recruitment and translation, but this reaction is somehow blocked during the editing process to avoid synthesis of aberrant proteins from mRNA lacking an open-reading frame[15]. To reconcile these observations, we envisaged that a *trans*-acting factor may recognize a nascent A-tail to enable an interaction between protein complexes occupying 5′ and 3′ mRNA termini. Consequentially, this would increase resistance to degradation and uridylation. In this study, we identified the PPR-containing KPAF4 as a factor essential for normal parasite growth and demonstrated its role in recognizing 3′ A-tails, preventing mRNA uridylation by RET1, and inhibiting degradation of adenylated mRNAs.

PPR proteins are defined by arrays of ~35-amino acid helix-turn-helix motifs[39], each recognizing a single nucleotide via amino acid side chains occupying cardinal positions 5 and 35[17]. Bioinformatic analysis of trypanosomal PPRs identified KPAF4 as a factor potentially capable of binding five consecutive adenosines, and, therefore, a candidate for a mitochondrial PABP. Biochemical fractionation, immunochemical, and proteomics experiments demonstrate that KPAF4 interacts with polyadenylation and RESC complexes. In agreement with an established architecture of the RESC, KPAF4 contacts are mostly confined to the PAMC, which has been defined as a docking site for the polyadenylation complex[25]. A binding platform for RNA editing substrates and products[3,40], RESC also recruits enzymatic RNA editing core complex and, importantly for mRNA

stabilization, the 5′ end-bound PPsome[4]. Therefore, it seems plausible that RESC-mediated interaction network provides a physical basis for functional coupling between 5′ pyrophosphate removal by MERS1, KPAP1-catalyzed 3′ adenylation, and internal editing events. To that end, in vivo crosslinking identified 3′ termini and short A-tails as KPAF4 primary recognition sites, but also detected binding to the 5′ region. KPAF4 CLAP-Seq coverage displayed an instructive correlation with the editing status: The 3′ termini including A-tails were occupied in all mRNA types (pre-edited, edited, and unedited), while the 5′ regions were bound chiefly in pre-edited mRNAs. Remarkably, these patterns were mirrored by editing-dependent re-distribution of MERS1-binding sites. Collectively, interaction networks, proximity studies, and identification of in vivo-binding sites point toward circularization as the mRNA surveillance and stabilization event. In this scenario, only adenylated pre-mRNA proceeds through the editing while being protected by KPAF4-bound short A-tail from the MPsome assault, which degrades RNA[5], and from A/U-tailing, which activates translation[15].

Although circularization is likely to take place in vivo, KPAF4 in vitro properties are also consistent with short A-tail-dependent inhibition of 3′–5′ degradation by the MPsome and 3′ uridylation by RET1 TUTase. Accordingly, the outcomes of KPAF4 knockdown revealed specific loss of A-tailed molecules, but minimal impact on post-editing A/U-tailing reaction, which is accomplished by KPAP1, RET1, and KPAF1/2 polyadenylation factors. It seems plausible that the A/U-tailed mRNA no longer depends on KPAF4-mediated stabilization mechanism. The argument can be extended to suggest that completion of editing results in KPAF4 displacement from the short A-tail and/or loss of interaction with the 5′ end. These events would enable RET1 access and trigger A/U-tailing. The presence of a protein sensor that monitors RNA-editing completion has been suggested[20], but further studies are required to decipher a signaling mechanism. The KPAF4 stabilizing role is somewhat similar to PPR10 in maize chloroplasts, which defines mRNA 3′ end by binding to a specific site and impeding 3′–5′ degradation[41]. The distinction in lies in post-trimming addition of the KPAP4-binding platform, the A-tail. Conversely, upregulation of some pre-edited (A6, CYB) and unedited (ND1) transcripts in KPAF4 knockdown also suggests a more nuanced transcript-specific functions of the PABP. These effects are reminiscent of the moderately destabilizing contribution of the A-tail structure[5,13], an unresolved phenomenon that requires further investigation.

In this example of convergent evolution, a PPR array in KPAF4 apparently carries a similar function to that of an RRM domain, a universal fold of canonical PABPs[42]. Although the recognition

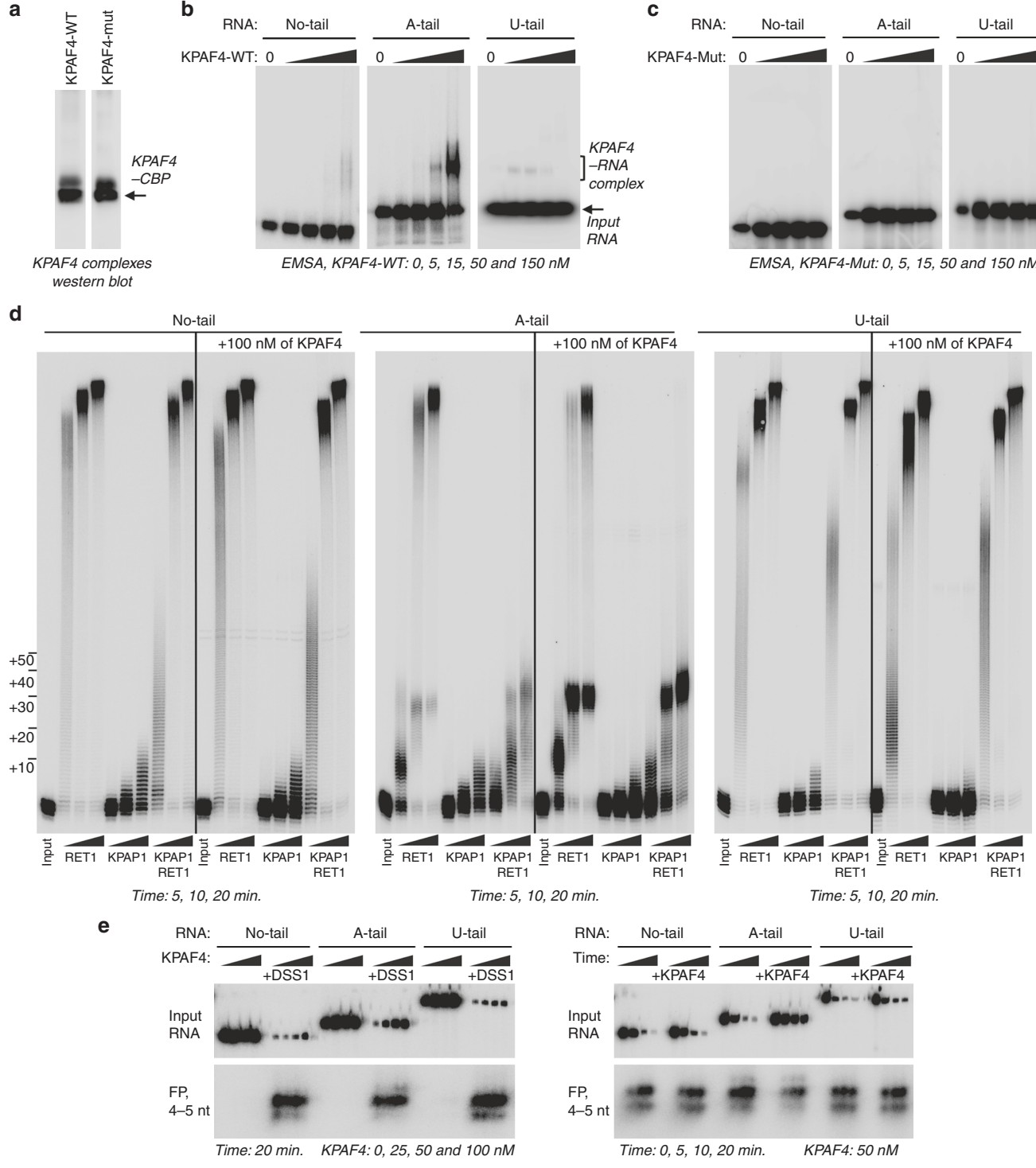

mechanisms are likely to be different, KPAF4 properties are well aligned with a paradigm for PPR repeats as sequence-specific readers and modulators of diverse enzymatic activities. The latter effects can be stimulatory, as typified by KPAF1/2[15] and KPAF3[5], or inhibitory, like those conferred by KPAF4.

## Methods

**RNA interference and protein expression**. Plasmids for RNAi knockdowns were generated by cloning an ~500-bp gene fragment into p2T7-177 vector for tetracycline-inducible expression[43]. Linearized constructs were transfected into a procyclic 29-13 *T. brucei* strain[22]. For inducible protein expression, full-length genes were cloned into pLew-MHTAP vector[44]. For BioID experiments, full-length genes were cloned into the same vector with the C-terminal TAP tag replaced by a mutated BirA* ligase from *E. coli*[33]. DNA oligonucleotides are listed in Supplementary Data 6.

**Biochemical analysis**. RNAi, mitochondrial isolation, glycerol gradient, native gel, total RNA isolation, northern, and western blotting, qRT-PCR, and tandem affinity purification were performed as described[45]. The change in relative abundance was calculated based on qRT-PCR, or northern blotting, data assuming the ratio between analyzed transcripts and control RNAs in mock-induced cells as 1% or 100%, respectively. Detailed protocol for BioID purification from crude mitochondrial fraction is provided in the Supplementary Methods.

**Fig. 7** KPAF4-bound adenylated RNA is partially resistant to uridylation and degradation in vitro. **a** Western blotting of affinity purified KPAF4-WT and KPAF4-Mut samples. Protein samples were purified from mitochondrial fraction by rapid affinity pulldown with IgG-coated magnetic beads. KPAF4 polypeptides were detected with an antibody against the calmodulin-binding peptide. Single experiment performed. **b** Electrophoretic mobility shift assay with KPAF4-WT. Increasing amounts of affinity-purified KPAF4 were incubated with 5′ radiolabeled RNAs and separated on 3–12% native PAGE. Representative of six experiments is shown. **c** Electrophoretic mobility shift assay with KPAF4-Mut was performed as in (**b**). Representative of two experiments is shown. **d** RNA adenylation and uridylation. KPAP1, RET1, or in combination, were incubated with 5′ radiolabeled RNA and ATP, UTP, or ATP/UTP mix, respectively, in the absence or presence of KPAF4. Recombinant enzymes were purified from bacteria as described[8,48]. Reactions were terminated at indicated time intervals and products were resolved on 10% polyacrylamide/ 8 M urea gel. Representative of seven experiments are shown. **e** RNA degradation. The same RNA substrates as in (**d**) were incubated with increasing concentrations of KPAF4, and the reactions were initiated by adding buffer or the MPsome for a fixed period of time (DSS1, left panels). RNAs were incubated with a fixed concentration of KPAF4 for 20 min, and reactions were initiated by adding the MPsome (right panel). Reactions were terminated at indicated time intervals and products were resolved on a 10% polyacrylamide/8 M urea gel. Input RNA and final degradation products of 4–5 nt (FP) are shown. Representative of two experiments are shown and quantified in Supplementary Fig. 4

**Protein identification by LC–MS/MS**. Affinity-purified complexes were sequentially digested with LysC peptidase and trypsin. LC–MS/MS was carried out by nanoflow reversed phase liquid chromatography (RPLC) (Eksigent, CA) coupled on-line to a Linear Ion Trap (LTQ)-Orbitrap mass spectrometer (Thermo-Electron Corp). A cycle of full FT scan mass spectrum ($m/z$ 350–1800, resolution of 60,000 at $m/z$ 400) was followed by 10 MS/MS spectra acquired in the LTQ with normalized collision energy (setting of 35%). Following automated data extraction, resultant peak lists for each LC–MS/MS experiment were submitted to Protein Prospector (UCSF) for database searching[46]. Each project was searched against a normal form concatenated with the random form of the *T. brucei* database (http://tritrypdb.org/tritrypdb/).

**Sequencing of RNA 3′ extensions**. Total RNA (10 μg) was circularized with T4 RNA ligase 1[8], digested with RNase R (Epicenter) to remove linear RNA, and termini were amplified with gene-specific primers listed in Supplementary Data 6. Two biological replicates of long-range single molecule real-time (SMRT) sequencing of 0.2–4 kb fragments was performed on a PacBio RS II system (Pacific Biosciences). Highly similar data sets were combined for final analysis. A single round of short-range sequencing was performed on a MiSeq instrument in 300 nt mode[47].

**Crosslinking-affinity purification and sequencing (CLAP-Seq)**. UV-crosslinking, affinity purification, and RNA-Seq library preparation from KPAF4-bound and MERS1-bound RNA fragments have been performed as described[45], with modifications outlined in Supplementary Methods.

**In vitro reconstitution**. Edited RPS12 mRNA fragments were prepared by in vitro transcription and 5′ radiolabeled.

No-tail: GGGTGGTGGTTTTGTTGATTTACCCGGTGTAAAGTATTATACACGTATTGU

AAGUUAGAUUUAGAUAUAAGAUAUGUUUUU

A-tail: GGGTGGTGGTTTTGTTGATTTACCCGGTGTAAAGTATTATACACGTATT

GUAAGUUAGAUUUAGAUAUAAGAUAUGUUUUUAAAAAAAAAAAAAAAAAAAA

U-tail: GGGTGGTGGTTTTGTTGATTTACCCGGTGTAAAGTATTATACACGTATTG

UAAGUUAGAUUUAGAUAUAAGAUAUGUUUUUUUUUUUUUUUUUUUUUUUUUU

MPsome assays were carried out in 20 μl reaction containing 50 mM Tris–HCl, pH 8.0, 1 mM DTT, 2 units/μl RNaseOut ribonuclease inhibitor (Life Technologies), 0.1 mM MgCl$_2$, 20,000 cpm of 5′-labeled RNA, 2 μl of TAP-purified DSS1 fraction, and 50 nM of KPAF4. The reaction was pre-incubated at 30 °C for 20 min, and started with the addition of DSS1. Aliquots were separated on 10% polyacrylamide/8 M urea denaturing gel. Phosphor images were acquired with Typhoon FLA 7000 (GE Healthcare).

**Reporting summary**. Further information on experimental design is available in the Nature Research Reporting Summary linked to this article.

## Data availability

All data generated or analyzed during this study are included in this article (and its Supplementary Information files). KPAF4 CLAP-Seq and tail sequencing data were deposited into the Sequence Read Archive (https://www.ncbi.nlm.nih.gov/sra) under accession number PRJNA477550. Custom sequence analysis scripts are available at www.tinyurl.com/y7x2txkh. The mass spectrometry proteomics data have been deposited to the ProteomeXchange Consortium via the PRIDE partner repository (https://www.ebi.ac.uk/pride/archive/) with the dataset identifier PXD012008. The Source Data underlying Figs. 1b–e, 3b–d, 4a, c–f, 6a, 7b–e, and Supplementary Figs. 1 and 4, are provided as a Source Data file.

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

## Acknowledgements

We thank members of our laboratories and Ruslan Aphasizhev for discussions and technical advice. This research was supported by NIH grant AI113157 to I.A.

## Author contributions

M.M., T.S., C.Y., F.M.S., and I.A. carried out the experiments and contributed to discussion. T.Y., L.Z., and L.H. analyzed data, and developed analytical tools and contributed to discussion. I.A. designed experiments and wrote the paper. I.A. serves as the guarantor.

## Additional information

**Competing interests:** The authors declare no competing interests.

