## [Peer Review File · Nature Communications]

Reviewers' Comments:

Reviewer #1:

Remarks to the Author:

A word-formatted version of this same text is attached to this review for easier reading.

Key results: This study addresses a clear hypothesis about how 3' ends of parasite mitochondrial mRNAs are protected during modification. A gap in the mechanistic pathway of regulation of intertwined post-transcriptional events was potentially filled. An understated key result of interest to the wider RNA community was that a PPR protein was unambiguously (to my mind) identified as a PABP. The outstanding features of the work are that different approaches to test the hypothesis were employed, including some elegant in vitro biochemistry, an important knowledge gap was filled, explanations for why each experiment were performed were provided, and results were nicely summarized in each results subsection. Clear writing. In my opinion, however, issues of scientific rigor, data sharing appropriate for today's standards, and an appropriate melding of the background, methods employed and the findings to the larger body of literature beyond that of the author laboratory and its partner group are not only needed, but would serve to enhance the relevance of the findings beyond that of addressing the explicit hypothesis. Finally, I have many concerns regarding data and methodology that currently make it almost impossible for me to analyze the work at the "big picture" level.

Validity: Assuming the many significant and minor concerns (below) have been addressed, the only potential underlying flaw that would potentially invalidate the individual conclusions of this study would be the time point at which studies involving the silencing of KPAF4 were performed, long after the observation of a growth phenotype (see below). Some large, overarching conclusions about this protein's consistent, global role in post-transcriptional processing are weak given contradictory data also present within this study. However, the more intermediate conclusions are solid.

Originality and significance: The work is impactful to the field because underlying mechanisms central to the coordination of post-transcriptional events must be established in order to understand how they can be perturbed and how they may be differentially utilized. The editing process consisting of insertion/deletion of uridine is specific to the mitochondrion of a group of protists (the kinetoplastids), and while most conclusions of this work are confined to this context, this unique type of editing has been a source of fascination to biologists of all stripes. The biochemistry of the KPAF4 PABP is a contribution that can extend beyond the field of parasitology. Is this the discovery of a new class of PABPs, or the first instance of a PPR that is specific to a homopolymer? If not, these examples should be cited when the KPAF4 is first introduced as a candidate. If KPAF4 is novel in this way, this should be stated and discussed because it argues for the study being published here rather than in a good specialist journal. The same goes for evidence that this protein that appears to bind to the 3' mRNA A-rich extensions also interacts with proteins found on the 5' end. Can authors place this new evidence of circularization of mRNA molecules in a discussion of circularization of mRNAs as a mechanism of protection in general (standard eukaryotic mRNAs come to mind). I'm curious whether this is a common motif in life.

Data & methodology:

Regarding reporting being sufficiently detailed and transparent to enable reproducing the results, see comments below under "Suggested improvements".

Concerns:

The use of PacBio sequencing to analyze individual sequences consisting of homopolymers highly concerns me. In my estimation, this is actually not critical to the bulk of the story and could be dropped from the complete study, in light of the work I think it would take to prove its adequacy for this application (possibly experiments involving spike-in of A and U homopolymer-containing RNA of known composition), and the unlikelihood that I think this will prove to be workable. If authors feel that long tail analysis must be performed, Illumina 10X may be an alternative.

However, both sequencing approaches to tail sequencing and analysis are deeply flawed as currently presented. This is because given the inherent qualitative differences between non-templated tail sequences (that cannot be mapped to references) and typical RNA-seq data that is typically acquired with these sequencing methodologies, universal methods for library generation and their analysis are not in place. Therefore, the investigator must either design a method de novo to analyze them, or else utilize a relatively recently developed tail analysis method. These tend to be specific to the species of RNA ends being analyzed and are often must require so much technical reporting that the tail analysis becomes itself a significant independent study (e.g. PMID 24582499, PMID 27445395, PMID 29106412). Here the authors have cited nothing, so I must assume their methods are novel and thus must absolutely positively be validated! For instance, verification that libraries (if involving PCR amplification) are not skewed by over-amplification, reporting of the quality of sequence output, reporting exactly how each step of processing is executed, reporting fraction of usable reads in final analysis vs raw reads obtained, and very importantly, verification that enough tails have been sampled to adequately capture tail diversity. This at a minimum would require more than one biological replicate during method development/validation. It appears that one 2016 published method is similar (and was developed on *T. brucei* mt tails) in which validation was performed, PMID26759453, but it is not cited. I could not find a PacBio example of deep tail analysis at all and none was referenced in this paper. It was also not clear from the Supplemental methods how PacBio and Illumina reads were processed to remove the 5' and 3' ends that are not adaptor sequence but are also not tail sequence (the extreme 5' and 3' ends of each molecule). Number of total tail sequences obtained, and fraction of them suitable for analysis for deep sequencing, were not provided.

Other Data and Methodology specific concerns:

172: It is stated that growth rate of knockdown was slowed after approximately 24 hrs already, which is a pretty strong phenotype. I believe the norm for *T. brucei* RNAi is to focus on the time points just prior to the functional result of knockdown being manifest at the level of growth. This is done in order to minimize the changes that observations are secondary or later effects. Why then, is only the first data point in Fig. 4 at 24 hrs, and further results shown at 2, 3, and 4 days? Can authors justify that the differences observed in RNA abundances in Fig. 3B at 55 hrs reflect anything meaningful? A profile of KPAF4 silencing level at these earlier time points is critical; the authors should report it. We also don't know at what time point following induction of silencing that the deep sequencing tail experiments were performed, so possibly those entire experiments are completely compromised.

Minicircle reference for iCLAP mapping was not provided. Also, were edited sequences catenated to the maxicircle sequence for mapping, or were these assessed independently? I do not know if the hits figure (6B) includes the edited hits. Also, authors noted that besides coverage at ends of mRNAs, there was also coverage throughout the 9S rRNA b/c it was abundant. I noted that coverage/bp, there are seemed to be just as high of coverage across ND9, RPS12, CR4 and ND3, even though I don't know if these are that abundant as a portion of the total RNA. What do the authors make of this?

Total number of unmapped vs mapped iCLAP reads was not provided.

Fig. 2 networks: The range of normalized spectral abundance values in included in the range from thinnest to thickest line edge width should be provided. Missing also is the meaning behind the line color differences. State what the unlabeled circle clusters in 2A represent. The clarity of the network diagrams would increase if circles representing the proteins tagged and pulled down, and those fused to BirA* were identified in the diagrams with a different color or different outline than the other proteins in the network.

The list of proteins that are considered contaminants and removed from LC-MS/MS experiments

described in the supplemental excel file should be included. I am assuming each tab in S1 and S2 tables labeled with a protein name are the list of proteins co-purified with that tagged entity. S1 Table seems to be missing tabs for GRBC1, L3, and S17, while there are tabs for KAF1 and MERS1 that I didn't think were pulled down in the experiment. Table is missing a tab for one of the five pulled down proteins (KPAF2).

Fig. 4: Why are the parent vector and KPAF3 or KPAF4 day 0, which should be identical, so different in amount of mRNAs detected? Why does pre-edited RPS12 probe detect so many huge precursors? Might this be genomic DNA? It seems to be right in the well.

323-324 and Fig 7: I assume the point of Fig 7A is to demonstrate similar abundance of WT and Mut KPAP2. However, this was already shown in S2A. What is missing to be able to evaluate this figure is total protein stained images of the protein preparations with size markers and authors' evaluation of its purity. Not sure whether this was the TAP- or TAP6his protein, or how stringently it was purified (are we evaluating purified KPAF4, or else a KPAF4-containing complex, especially considering that Fig. 1 suggests that it's a pretty "sticky" protein?). Likewise, no information is provided regarding the purity and methods to purify the MPsome, or KPAP1 and RET1 (are these recombinant proteins also used in previous published biochem assays?).

Fig 7: "no tail" RNA actually ends in a U5 tail. What length of nucleotides is considered a "tail"???? Really, these experiments are between a short vs long U tail transcript vs A-tailed transcript; indeed, a similar slight change in addition behaviour in the RET1 incubated rxns seen using both the U-tailed RNAs highlights that these substrates may actually be quite similar. At the very minimum this needs to be mentioned in the text.

Fig 7 D: There is no Time: 0 lane for each rxn should not be listed below each gel. Rather, there is an overall "Input RNA" lane. Nucleotides are not listed as rxn components in Supplemental Methods for this assay and should be added. Also, there are brackets present in the Figure that are never mentioned in the text. Finally, is 10 nM of KPAF4 sufficient to coat 50nM of RNA tails (substrate) in order to see its effect on tail extension? After all, you do not see a KPAF4/RNA complex in the EMSA until the 50mM KPAF4 concentration. Was an assay done to determine that 10mM was the concentration to use in the 7D assays? For that matter, what method was utilized to quantify the abundance of purified KPAF4?

335-336: This conclusion cannot be derived just from the information presented in 7D. If this conclusion is based off earlier work it should be cited.

337-338: I do not agree with authors' conclusion. At the earliest time point, products that have processed 10-20 nt in uridylation are if anything more abundant when KPAF4 is present. So clearly, KPAF4 is not protecting the A tail in such a way that it prevents the initiation and sustainment of uridylation to lengths seen on gRNAs and rRNAs. At later time points, the uridylation does terminate in smaller tails with the presence of KPAF4, but there appears to be a natural stall point at around 50?? nt of tail extension that is less able to be overcome when KPAF4 is present. I am not sure how that result fits into the overall hypothesis. I know this is just an in vitro assay, but might it not argue that the authors should at least consider whether any alternative hypotheses exist to explain their data? Size ladders on these 7D gels would also aid the reader in interpreting these results.

340-341: I do not see that KPAP1 enhances KPAF4 inhibition of uridylation. For this to be true, the level of inhibition between KPAF4 + and KPAF4- lanes would have to be greater in magnitude with RET1/KPAP1 than with RET1 alone. They look quite comparable to me. Directly comparing the RET1 vs RET1/KPAP1 rxns in the KPAF3-containing rxns alone is not the proper comparison because that

comparison is not properly controlled.

135-167: Since tagged proteins are inducibly expressed at a non-native locus for these experiments, authors should reveal how long they were induced prior to cell harvest, and whether or not any growth defects were apparent at time of harvest.

Data inconsistencies between experiments and previous studies to address:

Length of COI Illumina tails are quite inconsistent with those obtained using similar and earlier methods in multiple studies e.g. PMID17543398, 18464794, 29847594. One of these is the author's previous study.

Increases and decreases in abundance in Northern blots for pre-edited mRNAs do not agree with those of qRT-PCR (A6 pre and edited, and CYb edited). Which experiment is to be believed?

7:3 ratio of A to U in A/U translation-associated tails that has been repeatedly found in previous studies is not at all the ratio that is attained by PacBio sequencing of long tails, and it is completely not addressed.

Since the quality of results from PacBio for such data is highly doubtful, this difference should be taken seriously.

Is 12S sequence in 5B (that labeled as PacBio sequencing in the legend) actually just the Illumina sequencing in 5A? The values are identical. However the 12S tail base frequency in 5C and 5D are very different from each other, one profile which is completely consistent with what is reported in the text (5C), and one profile that has to be wrong (5D), and this is currently not addressed.

How come there are values for tail base frequency in the CO1 in the 1-40 range but tail percentages do not appear in the same range? I also believe that authors size-selected for PacBio sequencing, then are there really enough values in the 0-50 nt range that a tail base frequency can be assessed? I am not sure what is meant in the legend by "nucleotide percentage was calculated for each position that contained at least 5% of total extracted sequences". Then at the very maximum, couldn't the upper limit of positions you could analyze be 20 positions? Taken in total, something is not lining up or else it is not adequately explained how this was done...

Method inconsistencies:

Please explain exactly what is meant by "were analyzed" in line 139. I assumed it meant that those were the proteins pulled down to generate the interaction network. However, the Fig. 2 legend says that KPAP1, KAF2, KPAF3, GRBC1, MRP2, L3, S17, and Tb...2670 were pulled down.

It clearly says in the text that Illumina sequencing was used for unedited, pre-edited, and rRNA, while PacBio was used for edited sequences. 5B and 5D legends would have you believe that the unedited CO1 and 12S in B and D were done by PacBio. Which is it?

Appropriate use of statistics and treatment of uncertainties: Not currently up to journal standards. Please clarify whether error bars track either technical or biological replicates for qRT-PCR (3B). 3B analysis was, to my reading, practically the only instance in which the reproducibility of an experiment was explicitly stated. This MUST be remedied. Much of the analysis utilized gel electrophoresis where it is impossible to directly show multiple replicates. Where conclusions about differences in abundance are drawn from the experiment, authors should state that these are representative images of "x" number of experiments. For supplemental figures where changes in abundance are quantitated, error

bars would be nice. The cell growth experiment should have error bars. It is my experience that iCLAP experiments are usually performed one time, provided that correct controls are performed, but RNAseq analysis, particularly when using a new technique, utilizes biological replicate samples to understand how observed differences between conditions differs relative to normal sample to sample noise. Also, no statistical verification of any stated differences in the deep sequencing tail analyses were performed, so Fig. 5 results are merely descriptive results.

Conclusions: Despite the occasional and unaddressed conflicts in results obtained via different methods described below, I find the authors' conclusions at the end of most results sections to be appropriately restrained and accurate, albeit vague. I do find it strange that if the authors hypothesize KPAF4 to protect mRNAs from decay until they can be completely edited, that removal of it results in a much higher population of pre-mRNAs. A visual model may be necessary to explain this apparent paradox, that may involve interruption of a downstream process.

Suggested improvements:

I think the number of experiments performed already tests the limits of a single study; and I do not have suggestions for additional studies. Addressing "Data and Methodology" and "Statistics" concerns would improve the manuscript, including reporting KPAF4 silencing levels at least at the RNA level over time.

Nature Comm. currently has posted an editorial advocating for data sharing. Also, this is an NIH-funded study, thus carrying obligations for dissemination of raw genomic/transcriptomic/proteomic data. Missing are raw data from tail sequencing (manual and deep). The manually sequenced tails can be provided in FASTA or excel file, and raw tail deep sequencing data can be deposited in SRA just like standard RNAseq data. In fact, an example of *T. brucei* Illumina mitochondrial mRNA tail data already exists there (accession PRJNA296502). Even though this is not a "omics" journal, I think that at this level, in-house R-scripts are required to be made available (GitHub, etc.) or, if investigators are actually using modified versions of available R-scripts, they should be appropriately cited and the modifications described. Otherwise reproducibility cannot be assessed in the future.

Other minor:

S3 legend incorrectly references 7C rather than 7F.

S1 and S3 grey vs black lettering and gridlines are not consistent, at least when printed out.

Arrows in 1E that designate size ladders should be alongside each gel, not just on the far right, for easier comparison by the reader.

101: I assume that *Leishmania* KPAF4 homologue has the same T and N sites at each repeat? Just a few words to add that fact to the text would be good.

110: KPAP1

Why is cyb always lower case and italicized while the other mt mRNAs are upper case and not italicized?

158: proximity range for which BirA ligase can biotinylate would help readers less familiar w/ the technique place it into context.

Suppl. Methods: Cannot discern: in what experiments was the Tandem Affinity method of purification employed, and for which was the Rapid Affinity Method used.

Fig. 4D: revise order of blots so that it is consistent with the rest of them on the page (pre-edited vs edited).

Fig. 2A legend: KPAF4 was omitted from the list of pulled-down proteins.

What does the red highlighting mean on entries in the "cytoscape" tab of Table S2?

Although the heading "Parasite maintenance" is in the condensed and expanded Methods, maintenance of the organism is not described. Please say how these are growing. Also, one of the figure legends states that procyclic stage cells are used; but the text uses "actively respiring insect stage". Clarify that what is meant is that you are using procyclic cells, not that you are growing them in medium that forces them to respire more.

References: In my estimation this manuscript does not meet the bar in referencing previous literature or previous results appropriately, particularly other sources beyond their immediate lab groups, and is a primary flaw. Here are a few specific instances that could be remedied, although more effort in general to place this within existing knowledge.

26: I believe more recent reviews that could be cited have appeared since 2014, that also contain this information.

34: Cite this background information.

98-103: The predicted targeting of this protein was already analyzed and presented in tables in two previous studies, under its former gene number Tb10.6k15.0120, studies that came shortly after it was found that PPR proteins including this one were found in trypanosomes in a study in Arabidopsis. Interestingly, these studies came to opposite conclusions about predicted localization (nicely illustrating the ambiguity that the authors' resolve in this current study). Introducing the protein in the context of what was previously known about it in the primary literature (particularly since the protein is included in a tree showing sequence distances between T. brucei and L. major PPRs in one study) would be a courtesy to other investigators attempting to link information. Studies are PMID17646387 (as observed on KPAF4's Tritryp page) and PMID 16837079.

145-146: While it is true that the function of MRP1/2 is still not definitely resolved, progress has been made and I am not really sure why the (mostly older) citations were used here and not these others (at minimum): PMID 19620277, 15504736, 16730807, 18295767.

191: The old RPS12 reference should be replaced with a much more updated study of its editing and location of overlapping of gRNAs. PMID: 26908922

A discussion of results of tail analyses in terms of extensive U-composition in the short tails analyzed with Illumina sequencing was completely missing, given the fact that authors continue to call them poly(A) tails. They have been identified as containing other nucleotides, mostly Us, in multiple previous studies, and this should be cited I think, if the authors continue to include this data in this study.

References for where methods were derived, should be inserted where not present in the Supplementary Methods (Rapid Affinity Purification, BioID, library preps, iCLAP, tail studies unless methods are 100% novel). References for the identities of antibodies should be presented.

Clarity and context: The Abstract is clear and accessible to a broad audiences, but investigators should temper conclusions that were only inferred (e.g. it was not actually proven that translation control was affected by KPAF4 silencing, and others). The Introduction is definitely tailored to an audience already familiar with the large host of players in kinetoplastid mitochondrial post-transcriptional regulation but is well-organized and provides the fundamental connections necessary. The Discussion would benefit from any attempt to expand it beyond a summary of the major conclusions. Where might the field go next with this information? What is not yet known? What is the impact of this work beyond the field of parasitology? Most importantly, what are the limitations of this study?

Scope of expertise: While the mass spec analysis (Fig 2) seems quite rigorous in that five unique peptides are required for a relationship to be named as such, my expertise in this area is minimal. I am qualified to assess other areas.

Reviewer #2:

Remarks to the Author:

This manuscript describes a thorough and systematic analysis of the function of the protein KPAF4 in RNA editing in *Trypanosoma brucei*. This protein has not been studied in detail before to my knowledge. The manuscript shows convincingly that KPAF4 associates with the 3' end of pre-edited mRNAs, protecting them from degradation and from premature translation, as well as contributing to the circularisation of the protein-RNA complex prior to editing and translation. These conclusions are based on an impressive array of experiments including studies of RNA processing, protein-RNA and protein-protein interactions in wild-type cells and in cells deficient in KPAF4. I find the results clear and compelling and entirely plausible. These findings fill a gap in our understanding of how editing and translation are sequentially separated in this system, which is crucial to the synthesis of the correct protein sequences. This is very nice work and improves our knowledge of what was already by far the best-studied and best-understood insertional editing system. Conceptually it seems likely that all RNA editing systems may have a step that fulfils a similar function (preventing translation until editing is complete), but the details are likely to vary enormously and thus these results may not be directly applicable to other systems. Nevertheless, the general concepts and the methodological approaches used are broadly relevant to many researchers working on RNA processing.

Minor points

1. The initial choice to study KPAF4 was based on a prediction that it might bind poly(A) sequences, but this prediction (as presented) is not particularly convincing. The PPR motif alignment in Fig. 1A highlights the residues expected to hydrogen-bond to adenosine bases, but an unrealistic amount of leeway in the position of these residues has been allowed. Instead of using TPRpred (which the authors used), I used HMMER3 and the PPR motif models developed by Cheng et al (*The Plant Journal* 85:532-547 2016) and this approach delineates the first 5 motifs as follows:

```
SRWLTDELVALSKAGNWEAAISTFLQLQQANIVQSN  
VFHYTTVISACGRAGKWEAAMSIFDQMTKNEVKPN  
VYTYTAVINACASAEKADVALRMFAHMLADVPPN  
VQTMTALVNACARSGEWERAIKILRDCEELFVAPN  
VFTYTAAMDGCRRGGVWKPVDLLNEMRDPTRVRPN
```

Presented like this, each of the first 5 motifs has the TN 5th/last combination expected for A-recognising motifs. The authors could consider revising their figure to show the KPAF4 PPR motifs in this way, which fits better with their experimental results and may explain why the modifications they introduced (described in Supplemental Figures 2 and 3) didn't give the expected results.

2. The authors could give a nod to the work by Prikryl et al (*PNAS* 108: 415-420 2011) who were the first to show (to my knowledge) that a bound PPR protein can block 3'-5' exonuclease activity.

We sincerely thank the Reviewers for scrupulous manuscript analysis and valuable suggestions, which were most helpful in improving the content and presentation. In this revision, we attempted to broaden an appeal by polishing the narrative and figure legends while avoiding claims of novelty and adhering to a conservative interpretation of data. Requests for additional details were accommodated by including two new Supplementary Tables and one Supplementary Figure. Extra panels were also added to Fig. 1, Supplementary Figs. 3 and F4. Figure legends now reflect the number of independent experiments from which the representative was selected. Full-size image files for representative experiments and their quantitation are now collected in the Source Data File. We also updated a reference (Sement FM, S.T., Zhang L, Yu T, Huang L, Aphasizheva I, Aphasizhev R. Transcription initiation defines kinetoplast RNA boundaries) from bioRxiv to “PNAS, in press.” A near final version is still available at bioRxiv.

Reviewers' comments:

Reviewer #1 (Remarks to the Author):

Key results: This study addresses a clear hypothesis about how 3' ends of parasite mitochondrial mRNAs are protected during modification. A gap in the mechanistic pathway of regulation of intertwined post-transcriptional events was potentially filled. An understated key result of interest to the wider RNA community was that a PPR protein was unambiguously (to my mind) identified as a PABP. The outstanding features of the work are that different approaches to test the hypothesis were employed, including some elegant in vitro biochemistry, an important knowledge gap was filled, explanations for why each experiment were performed were provided, and results were nicely summarized in each results subsection. Clear writing. In my opinion, however, issues of scientific rigor, data sharing appropriate for today's standards, and an appropriate melding of the background, methods employed and the findings to the larger body of literature beyond that of the author laboratory and its partner group are not only needed, but would serve to enhance the relevance of the findings beyond that of addressing the explicit hypothesis. Finally, I have many concerns regarding data and methodology that currently make it almost impossible for me to analyze the work at the “big picture” level.

We appreciate the positive overall assessment of our work and thank the Reviewer for a thorough analysis, thoughtful criticism and constructive suggestions. We made an explicit effort to broaden the appeal and provide a more comprehensive context for our hypotheses and findings to the extent of manuscript size limits.

Validity: Assuming the many significant and minor concerns (below) have been addressed, the only potential underlying flaw that would potentially invalidate the individual conclusions of this study would be the time point at which studies involving the silencing of KPAF4 were performed, long after the observation of a growth phenotype (see below). Some large, overarching conclusions about this protein's consistent, global role in post-transcriptional processing are weak

given contradictory data also present within this study. However, the more intermediate conclusions are solid.

The Reviewer is undoubtedly aware that the inducible knockdown of an essential gene encoding a mitochondrial RNA binding protein provides a “window of opportunity” between the decline of a targeted polypeptide (0-48 hours), primary effects on mitochondrial RNAs that follow in 24-96 hours, and secondary effects typically appearing at later time points. Whereas the primary effects are instructive of the function, the secondary ones can be caused by the loss of membrane potential and ATP production, and ensuing cell death. With that in mind, we analyzed the impact of KPAF4 RNAi in a time-resolved manner in 24 hours intervals, e.g., Fig. 4. This is a well-accepted practice validated in many publications. To illuminate this point better, we added new panel A to Fig. 3 to demonstrate a timeline for KPAF4 mRNA downregulation by RNAi. The KPAF4 RNAi growth curve was repeated in triplicate and plotted with as mean with s.d. bars. This curve demonstrates that bifurcation of induced and uninduced curves becomes significant at ~72 hours, which is on par with typical moderate growth phenotype.

The single time points were used on two occasions: In Fig. 3 for qRT-PCR (55 hours of RNAi induction) and in Fig. 5 for tail sequencing (72 hours of RNA induction). We note that KPAF4 RNAi led to cell growth inhibition, not cell death. Fortunately, at both 55 and 72 hours the cells were still morphologically unaffected and displayed well preserved kDNA and membrane potential, as determined by DAPI and Mitotracker staining. We argue that including these rather mundane data will distract the reader more than benefit the content. Overall, we submit that such concerns would be warranted in a case of a stronger phenotype leading to cell death, such as KPAP1 poly(A) polymerase (PMID: 19070634, loss of membrane potential investigated and documented to appear at 96 hours), and/or at time points beyond 96 hours. However, in the case of KPAF4, the 55 and 72 hours end points are still within a time frame that permits observation of primary effects at the mitochondrial RNA level.

Originality and significance: The work is impactful to the field because underlying mechanisms central to the coordination of post-transcriptional events must be established in order to understand how they can be perturbed and how they may be differentially utilized. The editing process consisting of insertion/deletion of uridine is specific to the mitochondrion of a group of protists (the kinetoplastids), and while most conclusions of this work are confined to this context, this unique type of editing has been a source of fascination to biologists of all stripes. The biochemistry of the KPAF4 PABP is a contribution that can extend beyond the field of parasitology. Is this the discovery of a new class of PABPs, or the first instance of a PPR that is specific to a homopolymer? If not, these examples should be cited when the KPAF4 is first introduced as a candidate. If KPAF4 is novel in this way, this should be stated and discussed because it argues for the study being published here rather than in a good specialist journal. The same goes for evidence that this protein that appears to bind to the 3' mRNA A-rich extensions also interacts with proteins found on the 5' end. Can authors place this new evidence of circularization of mRNA molecules in a discussion of circularization of mRNAs as a mechanism of protection in general (standard eukaryotic mRNAs come to mind). I'm curious whether this is a common motif in life.

We agree that there are two main elements that warrant publication in a broad-scope journal, such as Nature Communications. Identification of a mitochondrial poly(A) binding protein

connects and rationalizes observations made by several laboratories and provides a critical mechanistic link between the well-studied mRNA polyadenylation, editing and 3'-5' degradation processes. Hopefully, this will be a significant contribution to understanding mitochondrial gene expression in Kinetoplastida. The broader implication for RNA biology rests with a discovery of PPR array's capacity to recognize a poly(A) tail, a textbook function thus far associated exclusively with highly conserved RRM domains. To our knowledge, this is a first report of a PPR protein acting as PABP, but we intentionally avoided any explicit claims to this effect. Such claims are subjective and may become incorrect before the paper is published. The stabilizing role of poly(A) binding protein-dependent mRNA circularization is not a new finding, but the specific mechanism revealed in our work is likely to be novel. Just like with PPR protein functioning as PABP, with mRNA circularization we have encountered an example of convergent evolution. Here, a NUDIX hydrolase MERS1 bound to the monophosphorylated mRNA 5' end mimics recognition of the cap structure in eukaryotic mRNAs by eIF4E. Again, we intentionally avoided claim of novelty, but appreciate the opportunity to discuss these questions here.

Data & methodology:

Regarding reporting being sufficiently detailed and transparent to enable reproducing the results, see comments below under "Suggested improvements".

Concerns:

The use of PacBio sequencing to analyze individual sequences consisting of homopolymers highly concerns me. In my estimation, this is actually not critical to the bulk of the story and could be dropped from the complete study, in light of the work I think it would take to prove its adequacy for this application (possibly experiments involving spike-in of A and U homopolymer-containing RNA of known composition), and the unlikelihood that I think this will prove to be workable. If authors feel that long tail analysis must be performed, Illumina 10X may be an alternative.

However, both sequencing approaches to tail sequencing and analysis are deeply flawed as currently presented. This is because given the inherent qualitative differences between non-templated tail sequences (that cannot be mapped to references) and typical RNA-seq data that is typically acquired with these sequencing methodologies, universal methods for library generation and their analysis are not in place. Therefore, the investigator must either design a method de novo to analyze them, or else utilize a relatively recently developed tail analysis method. These tend to be specific to the species of RNA ends being analyzed and are often must require so much technical reporting that the tail analysis becomes itself a significant independent study (e.g. PMID 24582499, PMID 27445395, PMID 29106412). Here the authors have cited nothing, so I must assume their methods are novel and thus must absolutely positively be validated! For instance, verification that libraries (if involving PCR amplification) are not skewed by over-amplification, reporting of the quality of sequence output, reporting exactly how each step of processing is executed, reporting fraction of usable reads in final analysis vs raw reads obtained, and very importantly, verification that enough tails have been sampled to adequately capture tail diversity. This at a minimum would require more than one biological replicate during method development/validation. It appears that one 2016 published method is similar (and was developed on *T. brucei* mt tails) in which validation was performed, PMID26759453, but it is not cited. I could not find a PacBio example of deep tail analysis at all and none was referenced in this paper. It was also not clear from the Supplemental methods how PacBio and Illumina reads were processed to remove the 5' and 3' ends that are not adaptor sequence but are also not tail sequence (the extreme 5' and 3' ends of each molecule). Number of total tail sequences obtained, and fraction of them suitable for analysis for deep sequencing, were not provided.

The Reviewer is correct in pointing out that no perfect method is available to sequence long (>50 nt) non-templated 3' additions. The sequencing by synthesis (Illumina platform) suffers from poor quality in homopolymer regions and short reads that require assembly. Hence, the two most popular Illumina-based methods, PAL-Seq and TAIL-Seq, are used to measure the length of the tail without determining the sequence or to sequence the beginning and the end of a tail, respectively. Like Illumina-based CircTAIL developed in the Zimmer lab and applied to trypanosomal mitochondrial mRNAs, our approach relies on well-established mRNA circularization – RT-PCR. We and others had used this method extensively and published on several occasions (PMID: 21474072, 18464794). In this work, our approach combines short tail sequencing on Illumina platform and single molecule real time (SMRT) sequencing of long tail on PacBio platform. Neither platform was able to satisfactorily accomplish both tasks since SMRT fares poorly on short fragments (less than 100 nt). SMRT is not a new technology and does not need to be validated; any claims to a novel method would be misleading. SMRT does bring a unique capacity to sequence the entire tail, which has several advantages: 1) Most importantly, it is a direct sequencing. No read assembly required, one read – one molecule; 2) No cloning – validates the bimodal tail structure without E. coli bias against long poly(A) sequences; 3) Very little, if any, sequence bias during polymerization reaction – reads through poly(A) regions. Still generates tens of thousands of reads per transcript for robust nucleotide composition and length distribution analysis; and 4) No length limit – we have been able to sequence tails up to 500 nt. Let's also consider that our objectives for tail sequencing were rather specific: To interrogate RNA length changes observed by northern blotting in Fig. 4. We only claim to have: 1) Discovered uridylation of A-tailed pre-edited mRNAs in KPAF4 RNAi background; 2) Confirmed the length and A/U composition of long tails in representative edited and unedited mRNA; 3) Confirmed lack of KPAF4 RNAi effect on ribosomal RNAs; and 4) Confirmed moderate stimulation of mRNA A/U-tailing upon KPAF4 repression. The nucleotide frequency in each position starting from encoded 3' end was the only parameter calculated and we are not sure what kind of validation would be required to corroborate our conclusions from direct sequencing.

To alleviate the Reviewer's concerns, we have added new Supplementary Figure 2 showing the quality of sequenced PacBio and Illumina libraries, and new Supplementary Table 4 detailing the read numbers and mapping statistics. It can be noticed that an increase in PCR product intensity (Fig. S2) correlates with qRT-PCR (Fig. 3B) and northern blotting (Fig. 4) data. For example, upregulation of pre-edited Cyb, RPS12 and A6 mRNA is reflected by more abundant PCR product. This indicates that libraries were not overamplified.

Other Data and Methodology specific concerns:

172: It is stated that growth rate of knockdown was slowed after approximately 24 hrs already, which is a pretty strong phenotype. I believe the norm for T. brucei RNAi is to focus on the time points just prior to the functional result of knockdown being manifest at the level of growth. This is done in order to minimize the changes that observations are secondary or later effects. Why then, is only the first data point in Fig. 4 at 24 hrs, and further results shown at 2, 3, and 4 days? Can authors justify that the differences observed in RNA abundances in Fig. 3B at 55 hrs reflect anything meaningful? A profile of KPAF4 silencing level at these earlier time points is critical; the authors should report it. We also don't know at what time point following induction of silencing

that the deep sequencing tail experiments were performed, so possibly those entire experiments are completely compromised.

This concern has been addressed in detail above. To reiterate, the 24 hours interval corresponds to approximately two cell divisions for procyclic parasites. This is a standard practice supported by many publications in top journals (PMID: 12954983, 18464794, 18951088, 21474072, 28684539, 16269544, 16285922 and 21252235, to name just a few). This time frame is accepted by the field for moderate growth inhibition phenotypes, which is the case with KPAF4. We have updated time points used for each experiment.

Minicircle reference for iCLAP mapping was not provided.

References have been updated to include:

1) Ochsenreiter, T., Cipriano, M. & Hajduk, S. L. KISS: the kinetoplastid RNA editing sequence search tool. RNA 13, 1-4 (2007).

2) Hong, M. & Simpson, L. Genomic organization of Trypanosoma brucei kinetoplast DNA minicircles. Protist 154, 265-279 (2003).

We also included non-redundant minicircle sequences deposited in GenBank.

Also, were edited sequences catenated to the maxicircle sequence for mapping, or were these assessed independently? I do not know if the hits figure (6B) includes the edited hits.

Maxicircle and edited sequences have been analyzed separately. However, reads mapping parameters were the same. Naturally, in the case of moderately edited mRNAs, such as Cyb or CO2, most reads mapped to both pre-edited and edited isoforms. Figure 6B shows only maxicircle mapping, as indicated. Fig. 6D shows an aggregated profile for pre-edited, edited and unedited transcripts because all these species are adenylated. Finally, Fig. 6E shows individual mRNAs, with their editing status indicated.

Also, authors noted that besides coverage at ends of mRNAs, there was also coverage throughout the 9S rRNA b/c it was abundant. I noted that coverage/bp, there are seemed to be just as high of coverage across ND9, RPS12, CR4 and ND3, even though I don't know if these are that abundant as a portion of the total RNA. What do the authors make of this?

Based on our prior experience with CLIP experiments for KPAF3 and guide RNA binding complex components, some ribosomal background seems inevitable. In aggregate, Fig. 6 makes a point that KPAF4 preferentially binds to mRNA 3' end, including the short A-tail.

Total number of unmapped vs mapped iCLAP reads was not provided.

Mapping statistics have been included in Supplementary Table 4.

Fig. 2 networks: The range of normalized spectral abundance values in included in the range from thinnest to thickest line edge width should be provided. Missing also is the meaning behind the line color differences. State what the unlabeled circle clusters in 2A represent. The clarity of the network diagrams would increase if circles representing the proteins tagged and pulled down, and those fused to BirA* were identified in the diagrams with a different color or different outline than the other proteins in the network. The list of proteins that are considered contaminants and removed from LC-MS/MS experiments described in the supplemental excel file should be included. I am assuming each tab in S1 and S2 tables labeled with a protein name are the list of proteins co-purified with that tagged entity. S1 Table seems to be missing tabs for GRBC1, L3, and S17, while there are tabs for KAF1 and MERS1 that I didn't think were pulled down in that experiment. Table is missing a tab for one of the five pulled down proteins (KPAF2).

Indeed, the networks in Fig. 2 should have been explained better. The NSAF range now has been provided in legends for Fig. 2 and the unnamed circle clusters have been labeled. The proteins that have been used for TAP purification and BioID are now marked by circles of different color. We have added extra sheets into Supplementary Tables 1 and 2 with raw unedited LC-MS/MS data. The legends and column names for Supplementary Table 1 have also been updated to show the origin of each target and corresponding NSAF values. There was an inadvertent mistake in listing tagged proteins in Fig. 2 legend, now corrected. Co-purification network was build based on affinity purification of KPAF4, its putative interactions partner Tb927.3.2670, MRP2, and known polyadenylation complex components (KPAP1, KPAF1, KPAF2 and KPAF3), and MERS1. The BioID proximity network included subunits from each complex of interest, MERS1 (PPsome complex), GRBC1 (RESC complex), KPAP1 (polyadenylation complex) and KPAF4. We thank the Reviewer for pointing out these discrepancies. The edges connecting KPAP1, MRP2, G2 an MERS1 were distinguished by color to overlay other links in the same area, visualization purposes only.

Fig. 4: Why are the parent vector and KPAF3 or KPAF4 day 0, which should be identical, so different in amount of mRNAs detected? Why does pre-edited RPS12 probe detect so many huge precursors? Might this be genomic DNA? It seems to be right in the well.

The parental transgenic 29-13 procyclic cell line bears two drug resistance genes while KPAF3 and KPAF4 RNAi cell lines grow under three drugs. It is not unusual for T. brucei cell lines grown under different selective markers to display somewhat different levels of individual mRNAs even though loading controls show a uniform amount of total RNA. With that in mind, adding RNA from a parental cell was just an extra control; the more rigorous reference is mock-induced RNAi, referred hereto as zero-time point. The probe for pre-edited RPS12 detects long (>1000 nt) precursors that are compressed at the top of 5% acrylamide gel. This is consistent with published data. The RNA samples have been DNase-treated and we'd appreciate some benefit of the doubt in our sample preparation techniques.

323-324 and Fig 7: I assume the point of Fig 7A is to demonstrate similar abundance of WT and Mut KPAP2. However, this was already shown in S2A. What is missing to be able to evaluate this figure is total protein stained images of the protein preparations with size markers and authors' evaluation of its purity. Not sure whether this was the TAP- or TAP6his protein, or how stringently it was purified (are we evaluating purified KPAF4, or else a KPAF4-containing complex, especially

considering that Fig. 1 suggests that it's a pretty "sticky" protein?). Likewise, no information is provided regarding the purity and methods to purify the MPsome, or KPAP1 and RET1 (are these recombinant proteins also used in previous published biochem assays?).

This is correct, western blotting in Fig. 7A demonstrates that affinity purified KPAF4 and KPAF4-mut fractions contain the same amount of KPAF4 protein. Conversely, Fig. S3A (numbering in the revised manuscript) shows western blotting in total cell lysates demonstrating tightly controlled expression at the same level. The KPAF4 proteins were purified differently for the enzymatic assay than for interaction network construction. Instead of conditions that preserve interactions (100 mM KCl, 5 mM MgCl₂), for binding and enzymatic assays we used conditions that strip most RNA-bound and weakly interacting proteins (RNase treatment, 0.5M NaCl, 5 mM EDTA, 1% MP40). This becomes apparent if one compares Supplementary Tables 1 and 5. Indeed, in the sample used for binding assays (Supplementary Table 5) we detected nearly stoichiometric amounts of KPAF4 and its putative binding partner, Tb927.3.2670, but not much more. The size of EMSA shift from an ~33 kDa RNA substrate fits well with a calculated apparent mass of KPAF4- Tb927.3.2670 heterodimer, which is ~60 kDa. We feel that LC-MS/MS analysis of KPAF4 and KPAF4-mut preparations is a more appropriate measure of sample composition than SDS PAGE protein profile. The Reviewer, however, has a legitimate point here – we should have provided more details about protein preparations used in experiments shown in Fig. 7. The Supplementary Information Appendix has been updated accordingly.

Fig 7: "no tail" RNA actually ends in a U5 tail. What length of nucleotides is considered a "tail"???? Really, these experiments are between a short vs long U tail transcript vs A-tailed transcript; indeed, a similar slight change in addition behaviour in the RET1 incubated rxns seen using both the U-tailed RNAs highlights that these substrates may actually be quite similar. At the very minimum this needs to be mentioned in the text.

The "no-tail" substrate was derived from encoded RPS12 mRNA sequence, which terminates with five Us. The "A-tail" RNA was generated by adding 20 As to the 3' end, which mimics short A-tails observed in vivo. The "U-tail" was a control in which 20As were replaced with 20 Us. We made no claim that "no tail" and "U-tail" substrates behave differently; apparently these five Us make no appreciable contribution. We do claim that in the presence of KPAF4, the "A-tail" RNA behaves differently from "no tail" and "U-tail" substrates in three respects: 1) It forms a specific complex with KPAF4; 2) It inhibits processive uridylation by RET1, an effect which is enhanced by KPAP1 presence; and 3) It inhibits RNA degradation by DSS1. We feel that these specific conclusions are well-supported by data shown in Fig. 7.

Fig 7 D: There is no Time: 0 lane for each rxn should not be listed below each gel. Rather, there is an overall "Input RNA" lane. Nucleotides are not listed as rxn components in Supplemental Methods for this assay and should be added. Also, there are brackets present in the Figure that are never mentioned in the text. Finally, is 10 nM of KPAF4 sufficient to coat 50nM of RNA tails (substrate) in order to see its effect on tail extension? After all, you do not see a KPAF4/RNA complex in the EMSA until the 50mM KPAF4 concentration. Was an assay done to determine that 10mM was the concentration to use in the 7D assays? For that matter, what method was utilized to quantify the abundance of purified KPAF4?

It appears that we erroneously omitted a decimal in KPAF4 concentration shown in panel 7D. This concentration was derived from EMSA experiment in Fig, 7B and should have been 100 nM, not 10 nM. The 100 nM concentration ensures binding of at least 50% of RNA substrate by KPAF4. We thank the Reviewer for this correction. The NTPs present in the reactions are now indicated in Supplementary Information.

335-336: This conclusion cannot be derived just from the information presented in 7D. If this conclusion is based off earlier work it should be cited.

The statement: “Adenylated RNA, on the other hand, caused the appearance of minor abortion products that were efficiently uridylated at later reaction time points” has been removed.

337-338: I do not agree with authors' conclusion. At the earliest time point, products that have processed 10-20 nt in uridylation are if anything more abundant when KPAF4 is present. So clearly, KPAF4 is not protecting the A tail in such a way that it prevents the initiation and sustainment of uridylation to lengths seen on gRNAs and rRNAs. At later time points, the uridylation does terminate in smaller tails with the presence of KPAF4, but there appears to be a natural stall point at around 50?? nt of tail extension that is less able to be overcome when KPAF4 is present. I am not sure how that result fits into the overall hypothesis. I know this is just an in vitro assay, but might it not argue that the authors should at least consider whether any alternative hypotheses exist to explain their data? Size ladders on these 7D gels would also aid the reader in interpreting these results.

Our statement that: “...KPAF4 inhibited RET1 activity on A-tailed RNA, and this effect was further enhanced by KPAP1” has been amended to “KPAF4 inhibited processive uridylation of A-tailed RNA by RET1 TUTase, and this effect was further enhanced by KPAP1.” This is an in vitro assay which corroborates the in vivo finding of A-tailed mRNA uridylation in KPAF4 RNAi background. We consider the revised statement to be sufficiently accurate.

340-341: I do not see that KPAP1 enhances KPAF4 inhibition of uridylation. For this to be true, the level of inhibition between KPAF4 + and KPAF4- lanes would have to be greater in magnitude with RET1/KPAP1 than with RET1 alone. They look quite comparable to me. Directly comparing the RET1 vs RET1/KPAP1 rxns in the KPAF3-containing rxns alone is not the proper comparison because that comparison is not properly controlled.

We'd like to stress that this experiment analyzes changes in RET1 processivity patterns depending on the nature of RNA substrate, and presence or absence of poly(A) polymerase and KPAF4, the proteins responsible for A-tail addition and A-tail binding, respectively. To this extent, the statement that KPAP1 selectively inhibits uridylation of “A-tail” substrate, and this effect is enhanced by KPAF4 are accurate. There is no KPAF3 in these experiments.

135-167: Since tagged proteins are inducibly expressed at a non-native locus for these experiments, authors should reveal how long they were induced prior to cell harvest, and whether or not any growth defects were apparent at time of harvest.

The Supplementary Methods were updated to indicate that protein expression was induced for 72 hours, at which point no appreciable growth phenotype has been observed (Supplementary Figure 3 B).

Data inconsistencies between experiments and previous studies to address:

Length of COI Illumina tails are quite inconsistent with those obtained using similar and earlier methods in multiple studies e.g. PMID17543398, 18464794, 29847594. One of these is the author's previous study.

Our previous study in 2011 used cloning of long tails in bacteria, which may not be the best method to assess the authentic length. The two other mentioned studies did not actually sequence tails. COI tail sequencing clearly illustrates the limitations of Illumina platform for sequencing of long tails, but still shows the beginning of a long A/U-tail. The approximately 400 nt range derived from SMRT sequencing matches the northern blotting estimates much closer.

Increases and decreases in abundance in Northern blots for pre-edited mRNAs do not agree with those of qRT-PCR (A6 pre and edited, and CYb edited). Which experiment is to be believed?

There is no discrepancy in the trend, i.e., what is declining by qRT-PCR is also declining by northern blotting. The two methods measure somewhat different populations: qRT-PCR detects all transcripts containing probe region, including decay products and precursors, while northern blotting detects the most prominent species. For this reason, in Supplementary Figure 1 provides graphs for individual mRNA isoforms detected by northern blotting in Fig. 4.

7:3 ratio of A to U in A/U translation-associated tails that has been repeatedly found in previous studies is not at all the ratio that is attained by PacBio sequencing of long tails, and it is completely not addressed. Since the quality of results from PacBio for such data is highly doubtful, this difference should be taken seriously.

The PacBio sequencing indeed revealed that the A/U ratio in long tails is closer to 50:50, than to 70:30 calculated in our original study. Again, two very different approaches (cloning and Sanger sequencing in PMID: 21474072 vs. SMRT in this study) produced closely matching results. The 70:30 ratio was derived from few cloned tails, while PacBio produced hundreds of thousands of single molecule reads, and eliminated cloning bias. We now mention this, and we are inclined to think of 50:50 ratio as more realistic. We are not sure what is meant by "results from PacBio for such data is highly doubtful" comment. To our knowledge, no conclusive data supporting functional implications of slightly different A/U ratios have been reported.

Is 12S sequence in 5B (that labeled as PacBio sequencing in the legend) actually just the Illumina sequencing in 5A? The values are identical. However the 12S tail base frequency in 5C and 5D

are very different from each other, one profile which is completely consistent with what is reported in the text (5C), and one profile that has to be wrong (5D), and this is currently not addressed.

Panels A and B show reads binning by length, and the results are consistent between the two platforms. Panels C and D show nucleotide frequency, and we made a specific point that SMRT sequencing is inapplicable for short reads, such as oligo-U tails on ribosomal RNAs, while Illumina clearly illustrates presence of U-tails, which is a well-established fact. In other words, PacBio counts positions in short reads well enough, but does sequence them well. Again, this is why applied both methods.

How come there are values for tail base frequency in the CO1 in the 1-40 range but tail percentages do not appear in the same range?

We are not sure what is the question here. Panels A and B provide a percentage of the total for each bin.

I also believe that authors size-selected for Pacbio sequencing, then are there really enough values in the 0-50 nt range that a tail base frequency can be assessed? I am not sure what is meant in the legend by “nucleotide percentage was calculated for each position that contained at least 5% of total extracted sequences”. Then at the very maximum, couldn't the upper limit of positions you could analyze be 20 positions? Taken in total, something is not lining up or else it is not adequately explained how this was done...

The PacBio libraries and the fragments excised for sequencing are now provided in Supplementary Figure 2A. The 5% cut off pertains to read support for the longest sequences considered in Fig. 5D.

Method inconsistencies:

Please explain exactly what is meant by “were analyzed” in line 139. I assumed it meant that those were the proteins pulled down to generate the interaction network. However, the Fig. 2 legend says that KPAP1, KAF2, KPAF3, GRBC1, MRP2, L3, S17, and Tb...2670 were pulled down.

The list of proteins has been corrected in response to previous comment and the bait proteins are now indicated in Fig. 2A. The method of analysis is stated in the previous sentence: “LC-MS/MS analysis of tandem affinity purified complexes....”

It clearly says in the text that Illumina sequencing was used for unedited, pre-edited, and rRNA, while PacBio was used for edited sequences. 5B and 5D legends would have you believe that the unedited CO1 and 12S in B and D were done by PacBio. Which is it?

We revised the corresponding Results section to make it clear that mRNAs expected have only short tails, such as pre-edited RPS12, A6 and cyb transcripts, we sequenced with Illumina while their edited forms known to have both short and long tails were sequenced with PacBio.

Unedited COI, expected to have short and long tails, and U-tailed rRNA were sequenced on both platforms.

Appropriate use of statistics and treatment of uncertainties:

Not currently up to journal standards. Please clarify whether error bars track either technical or biological replicates for qRT-PCR (3B).

Clarified in legend for Fig. 3B, biological replicates.

3B analysis was, to my reading, practically the only instance in which the reproducibility of an experiment was explicitly stated. This MUST be remedied. Much of the analysis utilized gel electrophoresis where it is impossible to directly show multiple replicates. Where conclusions about differences in abundance are drawn from the experiment, authors should state that these are representative images of “x” number of experiments. For supplemental figures where changes in abundance are quantitated, error bars would be nice. The cell growth experiment should have error bars. It is my experience that iCLAP experiments are usually performed one time, provided that correct controls are performed, but RNAseq analysis, particularly when using a new technique, utilizes biological replicate samples to understand how observed differences between conditions differs relative to normal sample to sample noise. Also, no statistical verification of any stated differences in the deep sequencing tail analyses were performed, so Fig. 5 results are merely descriptive results.

In addition to qRT-PCR, the only conclusions that rely on quantitation are those derived from northern blotting (Fig. 4 and Supplementary Figure 1). The changes in relative abundance were calculated against internal loading controls, such as nuclear encoded rRNA or tRNA, as appropriate for the size of mitochondrial RNA of interest. Growth phenotypes were counted against mock-induced culture maintained under the same conditions. When using automated cell counter, the replicates are simply unnecessary and rarely performed. Finally, we did make two technical replicates of PacBio libraries, but the sequencing results were virtually identical. Therefore, we combined the reads into one pool. We doubt that this level of technical details would benefit the paper, but it is now reflected in Supplementary Table 4.

Conclusions:

Despite the occasional and unaddressed conflicts in results obtained via different methods described below, I find the authors’ conclusions at the end of most results sections to be appropriately restrained and accurate, albeit vague. I do find it strange that if the authors hypothesize KPAF4 to protect mRNAs from decay until they can be completely edited, that removal of it results in a much higher population of pre-mRNAs. A visual model may be necessary to explain this apparent paradox, that may involve interruption of a downstream process.

The Reviewer brings a finer point here, which is still somewhat perplexing and cannot be resolved in this study. It pertains to the fact that short A-tail moderately destabilizes pre-edited mRNA (documented by Read and Aphasizhev labs), while it is absolutely required for edited mRNA stability. We have, to some degree, resolved the latter issue by demonstrating KPAF3 selective binding to pre-edited, but not edited mRNAs (Zhang et al, EMBO J, 2017). The former is unresolved and we are willing to speculate that it is not just the A-tail, but rather KPAF4-bound A-tail that, in concert with KPAF3 and perhaps KPAP1, moderately

destabilizes the pre-edited mRNA. We feel, however, that this issue requires further investigations and are not prepared to speculate any further in this paper.

Suggested improvements:

I think the number of experiments performed already tests the limits of a single study; and I do not have suggestions for additional studies. Addressing “Data and Methodology” and “Statistics” concerns would improve the manuscript, including reporting KPAF4 silencing levels at least at the RNA level over time.

We agree with this assessment.

Nature Comm. currently has posted an editorial advocating for data sharing. Also, this is an NIH-funded study, thus carrying obligations for dissemination of raw genomic/transcriptomic/proteomic data. Missing are raw data from tail sequencing (manual and deep). The manually sequenced tails can be provided in FASTA or excel file, and raw tail deep sequencing data can be deposited in SRA just like standard RNAseq data. In fact, an example of *T. brucei* Illumina mitochondrial mRNA tail data already exists there (accession PRJNA296502). Even though this is not a “omics” journal, I think that at this level, in-house R-scripts are required to be made available (GitHub, etc.) or, if investigators are actually using modified versions of available R-scripts, they should be appropriately cited and the modifications described. Otherwise reproducibility cannot be assessed in the future.

Data availability paragraph has been added.

The KPAF4 CLAP-Seq and tail sequencing raw data were deposited into the Sequence Read Archive (<https://www.ncbi.nlm.nih.gov/sra>) under PRJNA477550 accession number. Manually sequenced tails in pre-edited RPS12 mRNA have been compiled in a new Supplementary Table 4. Sequence analysis scripts are available at tinyurl.com/y7x2txkh.

Other minor:

S3 legend incorrectly references 7C rather than 7F.

Corrected to 7E.

S1 and S3 grey vs black lettering and gridlines are not consistent, at least when printed out.

Lettering and gridlines converted to black.

Arrows in 1E that designate size ladders should be alongside each gel, not just on the far right, for easier comparison by the reader.

We tried but the figure looks too crowded and unnecessary repetitions are distracting. These are pre-cast native gels, exceptionally reproducible. No changes have been made.

101: I assume that *Leishmania* KPAF4 homologue has the same T and N sites at each repeat? Just a few words to add that fact to the text would be good.

This is correct. Fig. 1A was redesigned based on Reviewer 2 suggestion to update repeat definition. The first five repeats in T. brucei and L. infantum KPAF4 are predicted to strictly recognize adenosine, which is now mentioned in the text.

110: KPAP1

Corrected.

Why is cyb always lower case and italicized while the other mt mRNAs are upper case and not italicized?

Made uniform with other mRNA names, upper case, not italicized.

158: proximity range for which BirA ligase can biotinylate would help readers less familiar w/ the technique place it into context.

An estimated range of 10 nm has been added along with reference to the original report (PMID24927568).

Suppl. Methods: Cannot discern: in what experiments was the Tandem Affinity method of purification employed, and for which was the Rapid Affinity Method used.

The protein purification methods have been re-arranged and clarifications have been made as to which methods were used for interaction network construction and in vitro reconstitutions.

Fig. 4D: revise order of blots so that it is consistent with the rest of them on the page (pre-edited vs edited).

Revised.

Fig. 2A legend: KPAF4 was omitted from the list of pulled-down proteins.

The figure legend was revised in response to earlier comment.

Although the heading "Parasite maintenance" is in the condensed and expanded Methods, maintenance of the organism is not described. Please say how these are growing. Also, one of the figure legends states that procyclic stage cells are used; but the text uses "actively respiring insect stage". Clarify that what is meant is that you are using procyclic cells, not that you are growing them in medium that forces them to respire more.

The growth conditions are now provided in Supplementary Information. The term "procyclic" is now used uniformly throughout the manuscript.

References: In my estimation this manuscript does not meet the bar in referencing previous literature or previous results appropriately, particularly other sources beyond their immediate lab

groups, and is a primary flaw. Here are a few specific instances that could be remedied, although more effort in general to place this within existing knowledge.

26: I believe more recent reviews that could be cited have appeared since 2014, that also contain this information.

A more recent 2016 review has been added.

34: Cite this background information.

Reference was added.

98-103: The predicted targeting of this protein was already analyzed and presented in tables in two previous studies, under its former gene number Tb10.6k15.0120, studies that came shortly after it was found that PPR proteins including this one were found in trypanosomes in a study in Arabidopsis. Interestingly, these studies came to opposite conclusions about predicted localization (nicely illustrating the ambiguity that the authors' resolve in this current study). Introducing the protein in the context of what was previously known about it in the primary literature (particularly since the protein is included in a tree showing sequence distances between T. brucei and L. major PPRs in one study) would be a courtesy to other investigators attempting to link information. Studies are PMID17646387 (as observed on KPAF4's Tritryp page) and PMID 16837079.

KPAF4 was predicted to be a PPR protein in previous surveys of trypanosomal PPRs, but its mitochondrial localization has not been investigated. Introducing more historical details in the Results section beyond those stated seems unnecessary.

145-146: While it is true that the function of MRP1/2 is still not definitely resolved, progress has been made and I am not really sure why the (mostly older) citations were used here and not these others (at minimum): PMID 19620277, 15504736, 16730807, 18295767.

More than a dozen studies notwithstanding, the MRP1/2 complex is still an enigma. We hope that reporting its in vivo proximity to polyadenylation and RESC complexes would guide future studies. This complex, however, is not the main subject here. Therefore, only the original discovery and some earlier key papers have been cited.

191: The old RPS12 reference should be replaced with a much more updated study of its editing and location of overlapping of gRNAs. PMID: 26908922

We prefer to use the original citations, as long as they have not been disproved by follow up studies.

A discussion of results of tail analyses in terms of extensive U-composition in the short tails analyzed with Illumina sequencing was completely missing, given the fact that authors continue to call them poly(A) tails. They have been identified as containing other nucleotides, mostly Us, in multiple previous studies, and this should be cited I think, if the authors continue to include this data in this study.

The short tail sequencing by Illumina is quite consistent with previous report including mostly As in RPS12 and A6, and a spike of Us before the A-tail in CYB. We think that contradiction would merit a lengthier discussion, but an agreement indicates that an adequate method was used here.

References for where methods were derived, should be inserted where not present in the Supplementary Methods (Rapid Affinity Purification, BioID, library preps, iCLAP, tail studies unless methods are 100% novel). References for the identities of antibodies should be presented.

The references have been inserted and the origin of antibodies clarified.

Clarity and context: The Abstract is clear and accessible to a broad audiences, but investigators should temper conclusions that were only inferred (e.g. it was not actually proven that translation control was affected by KPAF4 silencing, and others). The Introduction is definitely tailored to an audience already familiar with the large host of players in kinetoplastid mitochondrial post-transcriptional regulation but is well-organized and provides the fundamental connections necessary. The Discussion would benefit from any attempt to expand it beyond a summary of the major conclusions. Where might the field go next with this information? What is not yet known? What is the impact of this work beyond the field of parasitology? Most importantly, what are the limitations of this study?

These are all worthy questions and, admittedly, the Discussion is rather dry. Unfortunately, we are just 25 words short of the Nature Communication limit. Perhaps a follow up review would be a better place to expand the possible implications of our findings.

Scope of expertise: While the mass spec analysis (Fig 2) seems quite rigorous in that five unique peptides are required for a relationship to be named as such, my expertise in this area is minimal. I am qualified to assess other areas.

Reviewer #2 (Remarks to the Author):

This manuscript describes a thorough and systematic analysis of the function of the protein KPAF4 in RNA editing in *Trypanosoma brucei*. This protein has not been studied in detail before to my knowledge. The manuscript shows convincingly that KPAF4 associates with the 3' end of pre-edited mRNAs, protecting them from degradation and from premature translation, as well as contributing to the circularisation of the protein-RNA complex prior to editing and translation. These conclusions are based on an impressive array of experiments including studies of RNA processing, protein-RNA and protein-protein interactions in wild-type cells and in cells deficient in KPAF4. I find the results clear and compelling and entirely plausible. These findings fill a gap in our understanding of how editing and translation are sequentially separated in this system, which is crucial to the synthesis of the correct protein sequences. This is very nice work and improves our knowledge of what was already by far the best-studied and best-understood insertional editing system. Conceptually it seems likely that all RNA editing systems may have a step that fulfils a similar function (preventing translation until editing is complete), but the details are likely to vary enormously and thus these results may not be directly applicable to other systems. Nevertheless, the general concepts and the methodological approaches used are broadly relevant to many researchers working on RNA processing.

We appreciate a positive assessment of our work.

Minor points

1. The initial choice to study KPAF4 was based on a prediction that it might bind poly(A) sequences, but this prediction (as presented) is not particularly convincing. The PPR motif alignment in Fig. 1A highlights the residues expected to hydrogen-bond to adenosine bases, but an unrealistic amount of leeway in the position of these residues has been allowed. Instead of using TPRpred (which the authors used), I used HMMER3 and the PPR motif models developed by Cheng et al (The Plant Journal 85:532-547 2016) and this approach delineates the first 5 motifs as follows:

SRWLTDELVALSKAGNWEAAISTFLQLQQANIVQSN - 36
VFHYTTVISACGRAGKWEAAMSIFDQMTKNEVKPN - 35
VYTYTAVINACASAEKADVALRMFAHMRLADVPPN - 35
VQTMTALVNACARSGEWERAIKILRDCEELFVAPN - 35
VFTYTAAMDGCRRGGVWKPVDLLNEMRDPTRVRPN - 36

Presented like this, each of the first 5 motifs has the TN 5th/last combination expected for A-recognising motifs. The authors could consider revising their figure to show the KPAF4 PPR motifs in this way, which fits better with their experimental results and may explain why the modifications they introduced (described in Supplemental Figures 2 and 3) didn't give the expected results.

PPR definition has evolved since we discovered KPAF4 about five years ago, and the original text reflects the "logic of the day." However, the Reviewer is correct that the more recent definition by Cheng et al, which considers the fifth residue in the first helix and the last residue of the loop interconnecting adjacent motifs, is more robust. Indeed, it eliminates the need for variations and presents an even more compelling case for KPAF4 being a poly(A) binding protein by having five adjacent motifs that recognize adenosines. To make the presentation more rational than "historic," we re-designed Fig. 1 along with suggested repeat

structure and changed the associated text. The motif re-definition did not affect the amino acids mutated to confirm the specificity of KPAF4 binding in vivo and in vitro, except that including the last two repeats may not have been necessary. However, the mutated KPAF5 expresses at levels similar to WT, transitions into the mitochondrion and incorporates into the same complex. We are confident that KPAF4-mut is an adequate specificity control while dissecting contributions of individual repeats remains the work of the future.

2. The authors could give a nod to the work by Prikryl et al (PNAS 108: 415-420 2011) who were the first to show (to my knowledge) that a bound PPR protein can block 3'-5' exonuclease activity.

Done!

Reviewers' Comments:

Reviewer #1:

Remarks to the Author:

Only a few issues remain, but it is important that they are corrected prior to publication.

Time points of KPAF4 RNAi studies: RNAi KPAF4 abundance profile over time, inclusion of harvesting time for experiments, and a new growth curve largely addresses this concern, thank you. However, an additional small correction is now required. With additional replicates for the growth curve, there is now no growth phenotype until the 96 hour time point (3B graph), yet a growth defect by 24 hours is still described in the text (24 hours is also at odds with the response to reviewer in which it is characterized as a significant bifurcation of the curves at about 72 hrs). Fix text. Given the new growth curve, I completely agree that the time points utilized are appropriate.

Figure legend: Authors should actually state in the Fig. 2 legend that line colors have no values and are for visualization purposes only. The circles indicating the selected pulled-down proteins add great clarity, but one of them in Fig. 2a (MRP2) got bumped off a little and needs to be moved back.

Ambiguity in results from qPCR and Northern:

I found a potential mislabeling that may largely eliminate my concern. Is it possible that for 4C (A6 northern), the lane labeled "0" is actually the d(T) lane, the 24 hr lane is actually the 0 hr lane, and so on? d(T) is in the second position in some gels and the last position in others, so this could easily have happened during figure generation. Normally (as in all the other blots), d(T) + RNase H reduces the size of the long A/U band and the band tends to be a little fainter. The second lane looks like d(T) in the A6 gels.

Detailed Protocols: for antibodies for western blotting described as "in-house", authors should please cite the paper where they described them and add those citations to the reference list at the end of Detailed Protocols.

Conclusions: Authors did not want to address an unresolved issue in the text (increased abundance of pre-edited/not edited mRNAs in KPAF4 RNAi) because they did not want to speculate about it. Their desire to avoid speculation is completely legitimate. Would the authors consider just stating that this issue exists, but is currently unresolved and leave it at that?

189: replace "essential for parasite growth" with "essential for normal cellular function"???

228: add "the". 446: extra "in".

165-168: I see a brand-new paper has shed some light on MRP1/2 function and this context should perhaps be added (PMID 30120147).

Tail sequencing:

Please replace the Supplementary Figure 2 CO3p image that was apparently sequenced but never shown in Fig. 5 in this paper with the 12S that was shown (or just take CO3 out; 12S on a gel is not particularly informative). Likewise, either make some mention of the 9S, ND1, and CO3 sequences in the text (such as, "they yielded similar results so we did not show them or sequences were of too low abundance to use"), or else eliminate these primer sets and read abundance in information in the Supplementary Information and Table. It is confusing otherwise. Also, arrows are still not defined in the Fig. 5 legend. Please define or remove them.

The authors are correct in that they are not looking at the intimate details of tail sequences but rather

are simply trying to show broad characteristics of shortened poly(A) tail length and increased uridylation in KPAP4 RNAi cells, and thus, the quality of the Fig. 5 analysis only needs to be at the level that these points can be convincingly demonstrated. I think that this can largely be said of the short tails (Illumina) data, where this data does support the claim of "Discovery of uridylation of A-tailed pre-edited mRNAs in KPAP4 RNAi background" and "Confirmed lack of KPAP4 RNAi effect on ribosomal RNAs", as stated in the response to reviewers.

However, I was very surprised to see that only one Illumina sequencing replicate was performed. While high read depth for each tail population does mitigate this somewhat, this is highly unusual. Regardless of whether or not the authors have previously managed to publish Illumina sequencing analysis of single samples without replicates, the fact remains that there is no way to tell how much of observed difference is noise inherent in the unique PCR amplification and gel excision steps that do not occur in generation of standard RNA-seq library. Either publish replicate samples, or at a very minimum cite a study that addresses reproducibility of this sort of experiment that may partially justify your use of a single replicates for such tails, such as (PMID26759453). This should be noted in the text since the conclusion is based on increasing uridylation that is manifested differently in each of three transcripts, and on decreased poly(A) length that is a little hard to nail down because total tail length is measured, not length of the poly(A) region (therefore A6 pre-edited tails are greater in both the shortest AND longest bins). An assurance of likely reproducibility for such results seems important.

The authors have misunderstood my concerns regarding PacBio. I understand that SMRT itself needs no validation, but am very concerned about unknown degree to which PacBio's much higher error rate has on acquiring tail population length and composition that is actually reflective of the population, even if the likely issues cannot be identified a priori. It is not substitution but mainly insertion/deletion errors (potentially length-affecting) that are problematic, because of an inherent issue in the way the software recognizes the light signals. In sequences where the nucleotide diversity is reduced (2 nt rather than 4), the problem can be exacerbated. To utilize their PacBio reads, the authors should describe or cite papers to explain how they have accounted and/or corrected for error, as users of PacBio for traditional applications do. My guess is that this challenge is why nobody else has used PacBio for sequencing of low-complexity untemplated RNA yet (that I know of). I am only aware of correction algorithms that in some way require Illumina co-sequenced reads or a genomic reference. Theoretically, if authors have not utilized corrections, a sequencing of a pool of a few different known AU sequences of a variety of lengths and typical tail compositions could be performed, and the error for length and sequence determined, and a computational correction based on these results could be applied to the data. That would also validate its use as a diagnostic of sequence lengths and provide an estimate of the level of resolution that could be achieved with such an assay. There are hints in the presented data that these quality concerns are legitimate. The first is that (If I understand the processing and reported supplementary data correctly) only 9-39% (transcript-dependent) of trimmed reads are actually "clean tails" that are analyzed (Suppl Table). Since the only step between trimming and analysis is removing samples with less than 87% AU, that means that most of the tails had less than 87% AU composition, which is not a likely reflection of actual composition. When only 20% of the material you collect is usable, you have to ask what the bias might be introduced in what you throw away. Secondly, authors chalk up the mess in the first 60 nt of the PacBio compositional analysis in Fig. 5d to short reads of which PacBio cannot adequately distinguish composition. However, that problem also appears in the 1-60 region for CO1 reads, of which most are at least 100 nt long, so the failure of PacBio in reading the early parts of all these sequences is not really explained. The text describes these regions as "uninformative", when actually should have been explicitly described as "erratic portions of reads not reflective of the actual tail composition", and probably somehow indicated as such or removed entirely from the Fig. 5.

Furthermore, in assessing how well the claim of “moderate stimulation of AU tailing in KPAF4 repression” holds up even assuming error is irrelevant: I am not convinced. The subtle increases in tail abundances the authors note in the most common long tail sizes are paired with complementary subtle decreases in all the subsequent (longer length containing) bins for A6, RPS12, and COI (so actually long tails are shorter for these transcripts in KPAF4 RNAi), and the opposite may be happening for CYB. Compositional differences are more convincing, occurring in three of four tested transcripts. I am not sure why “confirming the length of long tails” would be necessary, as no one would dispute lengths determined from gel electrophoresis. In fact, I am not sure what the PacBio b. and d. parts of this figure contribute to the story of KPAF4 tail protection as quality control to avoid long tail addition until the completion of editing that is otherwise very succinctly laid out.

PacBio results do not APPEAR to be performed in biological duplicate as stated by the authors. In Supplementary Table 4, “Run 1” and “Run 2” capture tails from DIFFERENT transcripts. It is probably that four “runs” were performed and all numbers within Run 1 and Run 2 are the sum of two biological replicates. Please resolve this inadvertent ambiguity and then provide or summarize PCA analysis results that justify combination of datasets.

Finally, I just want an additional verification from the authors (authors have already kindly addressed this) that the bottom panels of both 5a (Illumina) and 5b (PacBio) are supposed to be totally completely identical for both WT and RNAi length partitions. The reason for asking again is because I see nt composition analysis for 12S in 5c (Illumina) out to about 18 nt, and in 5d (PacBio) out to about 70 nt. Since the methods stated that compositions are calculated for all positions with coverage of at least 5% of total extracted sequences, there are definitely some length differences of at least a subset of the reads. Therefore, I am thinking it may be mathematically unlikely to see these two graphs be completely identical and so I am confused.

Reviewer #2:

Remarks to the Author:

I'm happy with the revisions and the detailed responses to the points raised by the reviewers. I don't see any outstanding issues that need to be resolved.

Point-by-point response to Reviewer #1 Remarks

Only a few issues remain, but it is important that they are corrected prior to publication.

We truly appreciate the Reviewer's keen attention to details and helpful remarks.

Time points of KPAF4 RNAi studies: RNAi KPAF4 abundance profile over time, inclusion of harvesting time for experiments, and a new growth curve largely addresses this concern, thank you. However, an additional small correction is now required. With additional replicates for the growth curve, there is now no growth phenotype until the 96 hour time point (3B graph), yet a growth defect by 24 hours is still described in the text (24 hours is also at odds with the response to reviewer in which it is characterized as a significant bifurcation of the curves at about 72 hrs). Fix text. Given the new growth curve, I completely agree that the time points utilized are appropriate.

Corrected, the text now reads: "Inducible RNAi knockdown efficiently downregulated KPAF4 mRNA (Fig. 3a) and triggered a cell growth inhibition phenotype after approximately 72 hours, indicating that KPAF4 is essential for normal cellular function (Fig. 3b)."

Figure legend: Authors should actually state in the Fig. 2 legend that line colors have no values and are for visualization purposes only. The circles indicating the selected pulled-down proteins add great clarity, but one of them in Fig. 2a (MRP2) got bumped off a little and needs to be moved back.

Corrected. The legend for Fig. 2B now reads: "Edge colors other than red are for visualization purposes only." We also changed colors for circles indicating bait proteins in panel A to blue for consistency between panels A and B. The circle indicating MPR2 in panel A has been adjusted to a proper position.

Ambiguity in results from qPCR and Northern: s

I found a potential mislabeling that may largely eliminate my concern. Is it possible that for 4C (A6 northern), the lane labeled "0" is actually the d(T) lane, the 24 hr lane is actually the 0 hr lane, and so on? d(T) is in the second position in some gels and the last position in others, so this could easily have happened during figure generation. Normally (as in all the other blots), d(T) + RNase H reduces the size of the long A/U band and the band tends to be a little fainter. The second lane looks like d(T) in the A6 gels.

This is exactly what happened, our oversight. Apparently, in one of many revisions the designations were copied from panel E, where the oligo(dT) line is placed in the last lane. We routinely use approximately 2/3 of the normal RNA load in the oligo(dT) lane because the treatment typically causes adenylated RNA species to collapse into a more compact area. Lower intensity is useful to indicate the exact position of the

deadenylated RNA. The labelling has been corrected and all other panels were rechecked.

Detailed Protocols: for antibodies for western blotting described as “in-house”, authors should please cite the paper where they described them and add those citations to the reference list at the end of Detailed Protocols.

Agreed. Fortunately, all antibodies have been validated, published and disseminated among colleagues in the field. The references have been added for each antibody.

Conclusions: Authors did not want to address an unresolved issue in the text (increased abundance of pre-edited/not edited mRNAs in KPAF4 RNAi) because they did not want to speculate about it. Their desire to avoid speculation is completely legitimate. Would the authors consider just stating that this issue exists, but is currently unresolved and leave it at that?

This was indeed our sentiment, but in the revised version we ventured to indicate a plausible connection and stated that this phenomenon needs to be investigated further. The Discussion section now reads: “Conversely, upregulation of some pre-edited (A6, CYB) and unedited (ND1) transcripts in KPAF4 knockdown also suggests a more nuanced transcript-specific functions of the poly(A) binding protein. It seems plausible that such effects are reminiscent of the moderately-destabilizing contribution of the A-tail structure^{7,13}, an unresolved phenomenon that requires further investigation.”

189: replace “essential for parasite growth” with “essential for normal cellular function”???

Replaced to: “KPAF4 is essential for normal parasite growth and for maintaining a subset of mitochondrial mRNAs”

228: add “the”.

Added.

446: extra “in”.

Removed.

165-168: I see a brand-new paper has shed some light on MRP1/2 function and this context should perhaps be added (PMID 30120147).

The reference (#32) has been added in the following revised sentence: “A subject of extensive investigation, heterotetramer MRP1/2 RNA chaperone displays RNA annealing activity *in vitro*, but its definitive function remains undetermined²⁸⁻³¹, albeit contribution to mRNA stability seems likely³².”

Tail sequencing:

Please replace the Supplementary Figure 2 CO3p image that was apparently sequenced but never shown in Fig. 5 in this paper with the 12S that was shown (or just take CO3 out; 12S on a gel is not particularly informative). Likewise, either make some mention of the 9S, ND1, and CO3 sequences in the text (such as, “they yielded similar results so we did not show them or sequences were of too low abundance to use”), or else eliminate these primer sets and read abundance information in the Supplementary Information and Table. It is confusing otherwise. Also, arrows are still not defined in the Fig. 5 legend. Please define or remove them.

We eliminated CO3 panel in Supplementary Fig 2, which only served to illustrate the universally applicable cRT-PCR conditions. It is indeed a purely technical control. We eliminated 12S rRNA in Fig. 5 panels B and D (SMRT sequencing) because this RNA has only short U-tail, which is adequately read by Illumina. This is a meaningful control for mRNAs which also have short tails but composed of adenosines. As such, we use this example to illustrate the point that neither Illumina nor SMRT platforms are adequate to investigate both short and long tails in mRNA, but combining the two fulfills the objective (lines 263-267). The arrows in Fig. 5C and 5D are now explained as follows: “Arrows show positions of equal adenosine and uridine frequencies.” The purpose of these arrows is to draw reader’s attention to the fact that in KPAF4 knockdown uridylation of A-tailed mRNAs is more efficient. This is stated in the text as well.

The authors are correct in that they are not looking at the intimate details of tail sequences but rather are simply trying to show broad characteristics of shortened poly(A) tail length and increased uridylation in KPAP4 RNAi cells, and thus, the quality of the Fig. 5 analysis only needs to be at the level that these points can be convincingly demonstrated. I think that this can largely be said of the short tails (Illumina) data, where this data does support the claim of “Discovery of uridylation of A-tailed pre-edited mRNAs in KPAF4 RNAi background” and “Confirmed lack of KPAF4 RNAi effect on ribosomal RNAs”, as stated in the response to reviewers. However, I was very surprised to see that only one Illumina sequencing replicate was performed. While high read depth for each tail population does mitigate this somewhat, this is highly unusual. Regardless of whether or not the authors have previously managed to publish Illumina sequencing analysis of single samples without replicates, the fact remains that there is no way to tell how much of observed difference is noise inherent in the unique PCR amplification and gel excision steps that do not occur in generation of standard RNA-seq library. Either publish replicate samples, or at a very minimum cite a study that addresses reproducibility of this sort of experiment that may partially justify your use of a single replicate for such tails, such as (PMID26759453). This should be noted in the text since the conclusion is based on increasing uridylation that is manifested differently in each of three transcripts, and on decreased poly(A) length that is a little hard to nail down because total tail length is measured, not length of the poly(A) region (therefore A6 pre-edited tails are greater in both the shortest AND longest bins).

We agree with the Reviewer that the scope of our claim is limited to establishing increased uridylation in several adenylated transcripts, including pre-edited RPS12 and A6 mRNAs. We also thank the Reviewer for suggesting an appropriate reference to a study, which used somewhat different library preparation strategy to address the structure of 3' extensions in mitochondrial mRNAs. The rationale remains the same since no quantitative parameters have been determined, and there is an excellent agreement with Sanger sequencing shown in Fig. 4B. The PMID26759453 (reference#47) has been added to "Sequencing of RNA 3' extensions" paragraph in Methods.

The authors have misunderstood my concerns regarding PacBio. I understand that SMRT itself needs no validation, but am very concerned about unknown degree to which PacBio's much higher error rate has on acquiring tail population length and composition that is actually reflective of the population, even if the likely issues cannot be identified a priori. It is not substitution but mainly insertion/deletion errors (potentially length-affecting) that are problematic, because of an inherent issue in the way the software recognizes the light signals. In sequences where the nucleotide diversity is reduced (2 nt rather than 4), the problem can be exacerbated. To utilize their PacBio reads, the authors should describe or cite papers to explain how they have accounted and/or corrected for error, as users of PacBio for traditional applications do. My guess is that this challenge is why nobody else has used PacBio for sequencing of low-complexity untemplated RNA yet (that I know of). I am only aware of correction algorithms that in some way require Illumina co-sequenced reads or a genomic reference. Theoretically, if authors have not utilized corrections, a sequencing of a pool of a few different known AU sequences of a variety of lengths and typical tail compositions could be performed, and the error for length and sequence determined, and a computational correction based on these results could be applied to the data. That would also validate its use as a diagnostic of sequence lengths and provide an estimate of the level of resolution that could be achieved with such an assay. There are hints in the presented data that these quality concerns are legitimate. The first is that (if I understand the processing and reported supplementary data correctly) only 9-39% (transcript-dependent) of trimmed reads are actually "clean tails" that are analyzed (Suppl Table). Since the only step between trimming and analysis is removing samples with less than 87% AU, that means that most of the tails had less than 87% AU composition, which is not a likely reflection of actual composition. When only 20% of the material you collect is usable, you have to ask what the bias might be introduced in what you throw away. Secondly, authors chalk up the mess in the first 60 nt of the PacBio compositional analysis in Fig. 5d to short reads of which PacBio cannot adequately distinguish composition. However, that problem also appears in the 1-60 region for CO1 reads, of which most are at least 100 nt long, so the failure of PacBio in reading the early parts of all these sequences is not really explained. The text describes these regions as "uninformative", when actually should have been explicitly described as "erratic portions of reads not reflective of the actual tail composition", and probably somehow indicated as such or removed entirely from the Fig. 5.

The high error rate of PacBio (~15%) is a known problem, and in most genomics application the long (>20 kb) SMRT reads are used as templates for assembly of short Illumina reads. However, application of SMRT to short amplicons, such as those in our work (150-1000 bp), results in the same molecule being sequenced many times. PacBio refers to this application as circular consensus sequencing (CCS). In our runs, the primary reads exceeded 30 kb, which means that upon generation of CCS contigs each position was covered by at least 30 possesses. Therefore, the consensus base calling produced a high-quality final sequence (>99% accuracy PMID:19023044). In other words, according to our sequencing statistical data (tail length up to 400-450 nt for RPS12 ed., A6 ed., CO1 and up to 300-320 for Cyb ed.), ALL tail peaks on Fig. 5b with even barely noticeable difference between parental and KPAF4 RNAi cells reflect authentic difference between mRNA tail populations. The issue with reading short (<50 nt) tails is genuine and this necessitated a combination of PacBio and Illumina platforms, as we have explained previously. To avoid the argument on whether SMRT sequencing of short tails, such as those on 12S rRNA “uninformative” or, as suggested by the Reviewer “erratic portions of reads not reflective of the actual tail composition”, we eliminated 12S rRNA from SMRT panels B and D in Fig. 5, and left 12S rRNA in panels A and C intact. The latter show Illumina sequencing of short tails and are in full agreement with established presence of short U-tails in rRNAs.

Furthermore, in assessing how well the claim of “moderate stimulation of AU tailing in KPAF4 repression” holds up even assuming error is irrelevant: I am not convinced. The subtle increases in tail abundances the authors note in the most common long tail sizes are paired with complementary subtle decreases in all the subsequent (longer length containing) bins for A6, RPS12, and COI (so actually long tails are shorter for these transcripts in KPAF4 RNAi), and the opposite may be happening for CYB. Compositional differences are more convincing, occurring in three of four tested transcripts. I am not sure why “confirming the length of long tails” would be necessary, as no one would dispute lengths determined from gel electrophoresis. In fact, I am not sure what the PacBio b. and d. parts of this figure contribute to the story of KPAF4 tail protection as quality control to avoid long tail addition until the completion of editing that is otherwise very succinctly laid out.

To clarify the issue of potentially ambiguous evidence provided by relative tail length comparison in favor of “moderate stimulation of AU tailing in KPAF4 repression”, we would like to emphasize some of the statements arising from the text:

1) Reviewer’s proposition “so actually long tails are shorter for these transcripts in KPAF4 RNAi” does not actually contradict with A/U-tailing stimulation. We do not claim lengthening of A/U-tails.

2) We keep in mind that Fig. 5b represents relative changes in proportions of tails belongs to one transcript – Y axis: “Tails percentage, long range sequencing”. That “complementary subtle DECREASE” that Reviewer noticed “in all the subsequent (longer length containing) bins for A6, RPS12, and COI” is RELATIVE and cannot be

used to conclude about long tail shortening in KPAF4 RNAi. We did not use PacBio sequencing for absolute quantification between different datasets, but rather for relative measurements within a single dataset.

In our opinion, Fig. 5 b. supports the hypothesis about moderate stimulation of A/U-tailing under KPAF4 depletion. Again, we would like to provide Reviewer with a range of numbers used to build Fig. 5b plots:

1. Absolute amount of long tails (>150nt) in 2913 cells for RPS12 edited (10^5) for A6 edited (8×10^4) thousand, whereas in KPAF4 RNAi for the same transcripts the numbers were twice as high. Considering that the amount of starting material in control and experiment samples was the same, as well as number PCR cycles, we can assume that amount of long tails increase after KPAF4 knock-down.

2. The amount of long tails 300 to 400 nt range (can be considered as “a shrinking population”) in 2913 cells for RPS12 edited (3.6×10^3), for A6 edited (3.1×10^3), whereas in KPAF4 RNAi the number is 3.6×10^3 for both, so virtually the same in the control and in the experimental sample.

In conclusion, Fig. 5b demonstrates an increase in 151-300 nt-long tail population in ALL studied mRNA transcripts, which supports the claim of “moderate stimulation of AU tailing in KPAF4 repression”. These comparisons, not being bona fide quantitative, provide a support for our hypothesis, as stated in the text, not an ultimate proof. Perhaps one can consider a fact that this is chiefly a discovery work with much work ahead.

PacBio results do not APPEAR to be performed in biological duplicate as stated by the authors. In Supplementary Table 4, “Run 1” and “Run 2” capture tails from DIFFERENT transcripts. It is probably that four “runs” were performed and all numbers within Run 1 and Run 2 are the sum of two biological replicates. Please resolve this inadvertent ambiguity and then provide or summarize PCA analysis results that justify combination of datasets.

Run 1 and 2 captured some of the same, and some different transcripts. These two sets of the same transcripts were combined to increase read numbers. We are not sure what value PCA analysis would bring other than bury the reader in secondary details.

Finally, I just want an additional verification from the authors (authors have already kindly addressed this) that the bottom panels of both 5a (Illumina) and 5b (PacBio) are supposed to be totally completely identical for both WT and RNAi length partitions. The reason for asking again is because I see nt composition analysis for 12S in 5c (Illumina) out to about 18 nt, and in 5d (PacBio) out to about 70 nt. Since the methods stated that compositions are calculated for all positions with coverage of at least 5% of total extracted sequences, there are definitely some length differences of at least a subset of the reads. Therefore, I am thinking it may be mathematically unlikely to see these two graphs be completely identical and so I am confused.

As discussed in detail above, the 12S rRNA SMRT sequencing was eliminated from panels B and D in Fig. 5, and left panels A and C intact. These show Illumina sequencing of short tails, and are in full agreement with established presence of short U-tails in rRNAs in terms of their length and composition.

Reviewers' Comments:

Reviewer #1:

Remarks to the Author:

Thank you for your attention to these final details.